



# Radiative Transfer Model 3.0 integrated into the PALM model system 6.0

Pavel Krč[1], Jaroslav Resler[1], Matthias Sühring[2], Sebastian Schubert[3], Mohamed H. Salim[3,4], and Vladimír Fuka[5]

[1]Institute of Computer Science, Czech Academy of Sciences, Prague, Czech Republic
[2]Institute of Meteorology and Climatology, Leibniz University Hannover, Hannover, Germany
[3]Geography Department, Humboldt-Universität zu Berlin, Berlin, Germany
[4]Faculty of Energy Engineering, Aswan University, Aswan, Egypt
[5]Faculty of Mathematics and Physics, Charles University, Prague, Czech Republic

**Correspondence:** Pavel Krc (krc@cs.cas.cz)

**Abstract.** The *Radiative Transfer Model* (RTM) is an explicitly resolved three-dimensional multi-reflection radiation model integrated in the PALM modelling system. It is responsible for modelling of complex radiative interactions within the urban canopy and it represents a key component of modelling of energy processes inside the urban layer, and consequently PALM's ability to provide explicit simulations of urban canopy in meter-scale resolution. This paper describes RTM version 3.0 which

5   is integrated in PALM modelling system version 6.0. This version of RTM has been substantially improved over previous versions with new simulated processes, providing a more realistic representation of a wider range of urban scenarios, as well as with new discretization schemes and algorithms for a significantly better scalability and computational efficiency, enabling larger parallel simulations with up to many thousands of parallel processes.

## 1 Introduction

### 1.1 Overview of current solutions

Accurate representation of the spatio-temporal radiative exchange processes is essential for realistic modelling of the atmospheric boundary layer, especially with the urban boundary layer. These processes determine the energy budget of the surfaces and thus strongly affect boundary-layer dynamics as well as the spatio-temporal distribution of temperature, moisture and other scalar variables. In contrast to the synoptic-scale and mesoscale atmospheric models, microscale and building resolving

15   models encounter considerable challenges to accurately model such processes due to their fine spacial resolution as well as the heterogeneity of urban environments.

Mesoscale models typically limit modelling of radiation to the vertical direction, neglecting any horizontal radiative exchange. On the microscale, due to the complex shape of resolved obstacles (e.g. buildings, terrain or vegetation), the horizontal radiative interactions play an important role which can no longer be neglected and radiation should be modelled in all di-

20   rections, including shading and multiple reflections. This, however, increases the complexity of the numerical solution and





creates difficulties with respect to parallelization strategies via horizontal domain decomposition, which is commonly applied in atmospheric models (Tang et al., 2020), as the direct horizontal exchange is no longer limited to neighbouring subdomains.

Consequently, the method and the sophistication of modelling the radiative exchange within urban boundary layer vary in the microscale atmospheric models (Kim et al., 2020; Salim et al., 2018; Krayenhoff et al., 2014; Yang and Li, 2013; Franke et al., 2012; Gross, 2012; Früh et al., 2011; Heus et al., 2010; Huttner and Bruse, 2009; Krayenhoff and Voogt, 2007). They range from applying a simple parameterisation of radiative transfer, or even neglecting these processes altogether, to more explicit methods of radiative modelling. Moreover, some microscale models have limitations in simulating realistic urban domains. For example, some models use only the Reynolds-averaged Navier–Stokes equations (RANS) method for simulating the airflow, which is not always suitable for such applications. Also, some of these models are not designed to work on high performance supercomputers (HPC) with hundreds or thousands of CPU cores, which limits the size of modelled domains.

The RTM version 1.0 (Resler et al., 2017) was created in order to provide an open-source, HPC-enabled, fully 3-D model of radiative interactions inside the urban canopy, integrated into an urban climate model based on the large-eddy simulation (LES) method. Version 3.0 described in this paper provides substantial improvement over version 1.0 by including a wider selection of simulated processes for better representativity as well as featuring redesigned methods of discretization, improved algorithms and technical implementation for enhanced scalability and computational efficiency.

## 1.2  RTM role within the PALM model

Radiation processes are traditionally modelled in PALM by a one-dimensional radiation model with simulation of vertical radiation exchange without any lateral interactions (Maronga et al., 2015). The particular type of the radiation model can be selected and configured based on the requirements of the modelled scenario. The Rapid Radiative Transfer Model for Global models (RRTMG, see Clough et al., 2005) is available as well as a simpler clear-sky model (Maronga et al., 2020). Alternatively, users can prescribe a constant net radiation at the surfaces or use an external radiation input, such as observation data or a meteorological model output, for a better representation of cloud cover.

This one-column approach, however, is not sufficient to model the surface energy balance inside the urban canopy layer. This area is typically characterised by complex geometry of terrain, buildings and vegetation, where the radiative transfer processes in all directions cannot be neglected. Therefore PALM modelling system includes *Radiative Transfer Model* (RTM) as part of PALM-4U (PALM components for urban modelling). This model takes the radiation from the PALM radiation model, e.g. RRTMG, as input and calculates the radiation processes taking place inside the urban canopy layer explicitly in a fully 3-D geometry. By this, RTM provides the radiation fluxes as well as the surface net radiation including its components on the 3-D geometry, which are then used to model the surface energy balance, biometeorological quantities, etc. (see Section 4).

The main goals of the RTM development was to create a computationally efficient model which simulates all substantial radiative processes taking place inside urban canopy and which is fully integrated with the rest of the PALM model and its components, in particular:

- the spatial discretization of the domain matches the discretization of other parts of the PALM model,





– RTM is executed as part of the PALM program and it utilises its parallelization scheme and

– RTM utilises PALM data structures and subroutines as much as possible and provides its results directly back to PALM and its modules.

## 1.3 Changes since RTM version 1.0

The paper describes the current status of RTM implemtation in PALM, version 3.0, which is part of PALM version 6.0. This paper is a follow-up paper to Resler et al. (2017), which describes RTM version 1.0 as part of the *Urban Surface Model version*
*1.0* (PALM-USM) integrated in PALM version 4.0.

RTM version 1.0, as part of the PALM-USM module, has been evaluated with respect to performance and accuracy on a small urban scenario in Prague–Holešovice (Resler et al., 2017). The most important changes between RTM 1.0 and RTM 3.0 include:

– New discretization schemes for direct solar irradiance and for the sky view, which includes diffuse solar irradiance,
longwave irradiance from the sky and reflection as well as emission from surfaces towards the sky (see Section 2.4).

– New discretization scheme for the reflected and emitted radiation between surfaces (Section 2.2.4).

– Plant canopy interaction with LW radiation (absorption and emission, Section 2.3.2).

– Evapotransipration and latent heat flux in plant canopy (Section 4.3).

– Bidirectional integration with the radiation forcing model (e.g. RRTMG, Section 4.1).

– Calculation of mean radiant temperature for selected levels above ground with provision of radiant fluxes for the biometeorology module (Section 2.4.3).

– Integration of RTM within the PALM radiation module and coupling to all surface modules (Section 4.2).

– Multiple improvements, bugfixes and changes in interfaces with other PALM modules.

In order to quantify the differences brought by the new simulated processes and improved discretization schemes, a com-
parison study has been performed on the same scenario using PALM version 6.0 with RTM 3.0 with different sets of newly available simulated processes enabled. The results of this comparison are available in Krč (2019).

The paper is organized as follows: Section 2 describes the numerical approaches used in the RTM to consider the relevant physical radiation processes, while Section 3 describes the implementation of the RTM in PALM. Section 4 gives an overview
on how the RTM interconnects with other PALM modules. An evaluation of the RTM as well as a discussion of computational performance issues are presented in Section 5. Finally, Section 6 closes with a summary and ideas for further developments.





## 2    Numerical representation of radiative processes in RTM

The PALM model discretizes the modelled domain using a regular three-dimensional grid. The model supports arbitrary rotation of the grid along the vertical axis, with the default having the $x$-dimension representing the east–west axis in westward

direction and the $y$-dimension representing the south–north axis in northward direction. The $z$-dimension always represents the vertical axis in upward direction. Considering that the modelled domains are relatively small within the order of kilometres, PALM uses an f-plane approach with equidistant horizontal grid spacing rather than any spherical or ellipsoid geodetic projection.

Each of the horizontal $x$- and $y$-dimensions of the PALM grid is equidistant, so it is always regular. The vertical axis

may employ progressive *stretching*, which should be only applied well above the boundary layer, otherwise a bias could be introduced to the vertical turbulent transport. Also the RTM requires an equidistant grid inside the urban layer in which it operates.

For each horizontal coordinate $(x,y)$, PALM specifies a terrain height, which is discretized according to the vertical grid spacing, meaning that each grid box is either completely above or below the surface. When representing urban areas, the

obstacles include any permanent solid objects, e.g. buildings or terrain unevenness, while trees, shrubs and other resolved plant volumes are modelled separately as *plant canopy*. Terrain and buildings in RTM are currently limited to a so-called *2.5-D geometry*, which is able to represent radiative processes at upward- and horizontally facing surfaces and thus covers the majority of natural and large urban objects. Although PALM is also able to represent downward oriented surfaces, e.g. at bridges or overhanging parts of buildings, where its effect on the flow field is considered, RTM version 3.0 does not consider

them in radiative interactions yet. The model grid divides the 2.5-D surface geometry into individual surface grid elements which are referred to as *faces*. A face can be oriented northward, southward, westward, eastward or upward.

The RTM model considers two spectral ranges of electromagnetic radiation independently: the *shortwave* (SW) visible solar radiation and the *longwave* (LW) thermal radiation. The modelled radiation originates from the sun, the atmosphere, and all the modelled surfaces. The result of RTM is the amount of absorbed, reflected and emitted radiation for every face (both

horizontal and vertical) and the amount of absorbed and emitted radiation for each grid box containing resolved plant canopy (*plant canopy grid box*, PCGB). The model follows the radiation as it spreads from sources and as it propagates through the urban canopy layer and reflects off individual faces, taking into account model geometry, shading and mutual visibility between the faces, partial transparency/opacity of the plant canopy and reflective properties of the individual faces. Figure 1 gives an overview of the simulated processes. The detailed study of the contribution of the particular processes to the total simulated

radiative fluxes is described in Salim et al. (2020).

To limit the computational effort to a reasonable level, some less important processes have to be simplified or omitted entirely. These are:

**Infinite reflection.**  The model simulates only a finite number of reflections, after which the residual radiation is considered as

fully absorbed by the respective face it hits. This amount of radiation absorbed after the last reflection is also available





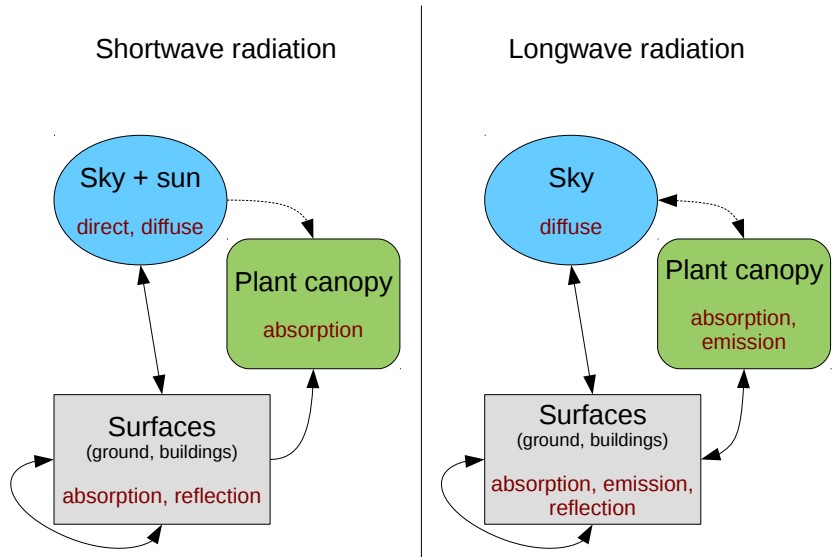

**Figure 1.** Radiative processes simulated by RTM version 3.0.

among the model outputs, allowing the model to be configured with an appropriate number of reflections so that the remaining amount is negligible. Depending on surface properties and model geometry, between 3 to 5 reflections are usually suitable for real-world urban scenarios (see Section 5.3).

**Directional reflection.** The current version of the model only supports diffuse (non-directional) reflection, specifically, all
surfaces are considered as Lambertian reflectors. Modelling specular reflection is planned for later versions of the model to better simulate glass and polished surfaces, which are also present among typical urban surfaces. However, this feature depends on the addition of arbitrarily oriented faces to PALM in order to realistically utilize its effect. Arbitrarily oriented faces are planned for the next major revision of PALM.

**Beam interaction with air.** Neither absorption, scattering, nor thermal emission by air mass is modelled inside RTM, con-
sidering rather short ray paths in the urban canopy layer. Even though the RRTMG includes these processes in a one-dimensional framework, any modifications within the urban canopy layer are not considered. In particular for fog, dense smog or heavy precipitation events this may become relevant, making RTM less suitable for simulating such scenarios.

**Some aspects of plant canopy interaction.** Modelling of plant canopy within RTM focuses mainly on its effects on other surfaces and on overall energy balance. In order to reduce computational complexity, it simplifies or disregards some
processes which are more relevant to fluxes within the plant canopy itself: reflection from plant canopy and internal interactions (emission and absorption among adjacent PCGBs). The reasons and impacts are discussed in Section 2.3.2.





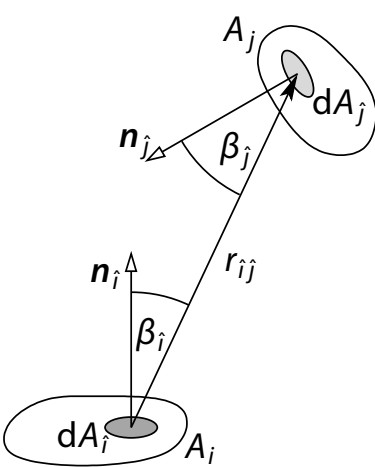

**Figure 2.** Calculation of the view factor between surface $A_i$ and $A_j$ by integrating over $A_i$ and $A_j$. $\boldsymbol{n}_{\hat{i}}$ and $\boldsymbol{n}_{\hat{j}}$ are the respective normal vectors of the surface elements $\mathrm{d}A_{\hat{i}}$ and $\mathrm{d}A_{\hat{j}}$, which are $r_{\hat{i}\hat{j}}$ apart.

**Thermal capacity of plant leaves.** Typical plant leaves are thin and lightweight, having a very small thermal capacity with respect to their surface area. This leads to rapid equalisation of their temperature with the temperature of the surrounding air via sensible and latent heat flux. In RTM, the simulated thermal capacity of plant leaves is zero and their temperature is identical to that of the surrounding air. This means that the heat flux of absorbed and emitted radiation is directly applied to the air mass. This approach is common in the field of atmospheric modelling (see e.g. Dai et al., 2003).

## 2.1 Representing radiative interactions using view factors

The discretization of RTM uses the same Cartesian grid as the rest of the PALM model. Each radiative quantity is modelled as a singular value per surface discretization unit (face) and the propagation of radiation is described as interactions between mutually visible faces.

The model considers all reflections and emissions to be Lambertian (i.e. ideally diffuse), following the Lambert's cosine law where the amount of radiation leaving the surface in one direction is proportional to the cosine of the angle $\theta$ between that direction and the surface normal. The interaction between faces can therefore be described similarly for reflection and for thermal emission.

For any two mutually visible faces $i$ and $j$, the *view factor* (VF) $F_{i \to j}$ is the fraction between the radiant flux originating from face $i$ that strikes face $j$ and the total radiant flux leaving face $i$. In an enclosed system where all radiative transfer happens between faces $1, \ldots, n$, the energy is conserved and the sum of all view factors from each particular face $i$ equals one:

$$\sum_{m=1}^{n} F_{i \to m} = 1 \,. \tag{1}$$





The value of $F_{i \to j}$ is calculated by integrating over the areas $A_i$ and $A_j$ (see Figure 2):

$$F_{i \to j} = \frac{1}{A_i} \int\limits_{A_i} \int\limits_{A_j} \frac{\cos \beta_{\hat{i}} \cos \beta_{\hat{j}}}{\pi r_{\hat{i}\hat{j}}^2} \, dA_{\hat{j}} \, dA_{\hat{i}} \,. \tag{2}$$

Here, $r_{\hat{i}\hat{j}}$ is the distance between the surface elements $dA_{\hat{i}}$ and $dA_{\hat{j}}$. $\beta_{\hat{i}}$ and $\beta_{\hat{j}}$ are the angles between the normal vectors of the respective surface element and their connection. Note that the integral is symmetrical for faces $i$ and $j$, which leads to the reciprocity property $F_{i \to j} A_i = F_{j \to i} A_j$. Applying that to (1) we get for each *target* face $i$

$$\sum_{m=1}^{n} \frac{F_{m \to i} A_m}{A_i} = 1 \,. \tag{3}$$

This formula allows to describe the face $i$, now being considered as the target for incoming radiation, as an observer whose field of view is a sum of portions. Each portion $\frac{F_{j \to i} A_j}{A_i}$ represents the view in the direction of a specific source face $j$, while the size of that portion takes into account the respective solid angle and the cosine law. The fraction $\frac{F_{j \to i} A_j}{A_i}$ is further called the *irradiance factor* $j \to i$, because it can be used to calculate the total irradiance $E_i$ of face $i$ using known radiosities of other faces and the irradiance factor values, which are equal to view factor values in the opposite direction:

$$E_i = \frac{\Phi_i^E}{A_i} = \frac{\sum_{m=1}^{n} \Phi_m^J F_{m \to i}}{A_i} = \frac{\sum_{m=1}^{n} J_m A_m F_{m \to i}}{A_i} = \sum_{m=1}^{n} J_m F_{i \to m} \,,$$

where $\Phi_i^E$ is total radiant flux received by the target face $i$, $\Phi_m^J$ is the radiant flux leaving the source face $m$ and $J_m$ is the radiosity of the face $m$. It can be seen that the irradiance of face $i$ is the weighted average of radiosities of other faces where the weights are equal to the irradiance factors.

### 2.1.1 Precalculated view factor values

The view factors values carry all the information about the geometry of the urban layer necessary for calculating propagation of reflected and emitted light among surfaces. Once they are known, calculation of the instantaneous fluxes can be reduced to simple vector multiplication. Determining the view factor values consists of multiple steps:

1. Establishing mutual orientation and position. In the rectangular grid, this is a matter of performing multiple coordinate comparisons to find out whether, for each face, the other face lies in the half-space above the plane of the first face, i.e. whether its angle $\theta$ is less than $\frac{\pi}{2}$.

2. Determining obstacles on the ray path between the faces. The obstacles may be fully opaque (terrain, buildings) or partially transparent, in which case a fraction of the radiant flux between the faces is absorbed. In RTM, the only partially transparent obstacle is the grid-resolved plant canopy, which is represented as a 3-D field of *leaf area density* (LAD). The fraction of the radiant flux allowed to pass through the obstacle and the radiant flux carried by the ray upon striking the obstacle is called *transmittance*. For plant canopy, it depends on the length of the ray's intersection with the respective PCGB, the LAD value at that PCGB and the extinction coefficient.





3. Calculating the actual view factor value. Although the exact value for two rectangular faces could be solved as a quadruple integral for each point of each of the faces, RTM uses simplifications in accordance with discretization of other processes in order to avoid the exact calculation.

The second step is implemented in RTM using a *raytracing* algorithm. This process is computationally complex, as it performs calculations involving each grid box that each traced ray intersects and it also can cause very high demands on the interprocess communication (see Sections 2.2.2 and 2.2.6). In PALM, each parallel process is responsible for modelling a horizontally divided *subdomain* within the modelled domain and most of the data stored locally is limited to the extent of the subdomain. Depending on the provided interconnecting infrastructure and the *Message Passing Interface* (MPI) implementation, the access to the values in other subdomain carried by MPI interprocess communication may be significantly slower than similar local memory access. Depending on the domain size and geometry, each traced ray may cross many subdomains. The complexity of this processing is further examined in Sections 2.2.2 and 2.2.6.

Due to this complexity, the raytracing task takes place during the model initialization phase before the actual simulation of time-steps begins. The values representing the view factors and other relevant data are precomputed, exchanged among the parallel processes and stored in such a way that the amount of calculations and MPI communications performed during computation of time-steps is minimized.

## 2.2 Discretization of the view in RTM

RTM version 3.0 offers two selectable methods for simulation of the irradiance of each face by providing two different schemes for discretization of the view from each face, which is represented by a set of irradiance factors. The legacy discretization scheme (originally introduced in RTM version 1.0, see Resler et al. (2017)) simulates the view from each target face as a set of irradiance factors from the centre of each face that is visible from the target face's centre. This leads to the requirement of performing raytracing between each pair of mutually visible faces and to the worst-case complexity of $\mathcal{O}(n^4)$ with respect to domain size as is described later in Section 2.2.2.

The current angular discretization scheme uses a different simplification with a better trade-off between complexity and accuracy and a guaranteed worst-case total number of view factors of $\mathcal{O}(n^2)$ (see Section 2.2.6). It also uses a newly developed 2D-raytracing algorithm which is optimized with respect to CPU time, memory consumption, and MPI interprocess communication by utilizing the geometric properties of the discretization scheme.

### 2.2.1 Legacy discretization of the view

While establishing the mutual visibility between two faces, the path between the faces is represented by a single ray connecting their centres. This is in accordance with the general principle of discretization by a rectangular grid, where the area or volume covered by each face or grid box is represented by a single scalar value and where the resolution can be increased if more spatial precision is needed. This way only one ray needs to be traced for every two faces oriented towards each other, and mutual visibility (absence of shading by an intermediate solid obstacle, i.e. building or terrain) becomes a binary relation. In



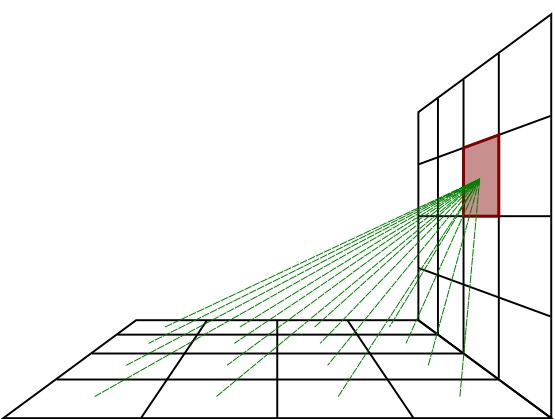

**Figure 3.** Illustration of the legacy view discretization scheme. For the highlighted face, raytracing is performed between its centre and the centres of its visible faces, creating a set of its view factors.

reality, however, any face may be illuminated only on part of its area and each target point illuminated by a non-point light
source may lie in penumbra of an obstacle. For the purpose of calculating the radiant flux absorbed by semi-opaque objects, it is assumed that the single ray between face centres carries the whole radiant flux leaving the source face towards the target face.

Together with simplifying the raytracing, also the view factor value calculation is simplified. Instead of solving the full integral in (2), the value of the integrand at the centres $C_i$ and $C_j$ of faces $i$ and $j$, respectively, is used to estimate the full
integral. With this, the approximate view factor $\tilde{F}_{1 \to 2}$ is given by

$$\frac{F_{i \to j}}{A_j} \approx \frac{\tilde{F}_{i \to j}}{A_j} = \frac{1}{A_j} \frac{1}{A_i} \frac{\cos \beta_{C_i} \cos \beta_{C_j}}{\pi r_{C_i,C_j}^2} A_i A_j = \frac{\cos \beta_{C_i} \cos \beta_{C_j}}{\pi r_{C_i,C_j}^2} . \tag{4}$$

The induced error is smaller for very distant faces and larger for faces close to each other. This error is considered acceptable within the resolution of the model, as it can always be reduced by increasing the resolution. The more important issue of this approach is that the sum of the approximate view factor values is no longer guaranteed to equal one. Because of this,
the modelled system could artificially gain or lose energy and possibly even diverge exponentially in time. To guarantee the conservation of energy, the normalization of the approximate view factor values is used in order to maintain (3) and the normalized view factor $\hat{F}$ is thus calculated by:

$$\frac{\hat{F}_{i \to j}}{A_j} = \frac{\frac{\tilde{F}_{i \to j}}{A_j}}{\sum_{m=1}^{n} \frac{\tilde{F}_{m \to j}}{A_j} A_m}$$





### 2.2.2 Computational complexity of the legacy discretization

The asymptotic complexity and scalability of the RTM can be evaluated using two different approaches: considering either a domain growing in size horizontally, while the vertical size and typical shapes of obstacles are kept constant, or considering a gradually increasing resolution for the same domain, which increases the amount of discretized data in each dimension.

The complexity and scalability for the latter case can be determined exactly. The number of faces increases proportionally with the surface area. For a horizontal domain size of $i \times j$ grid points where the resolution is increased by a factor $\varphi$ to $\varphi i \times \varphi j$

grid points, the number of faces grows by $\varphi^2$ and the number of other faces, to which each face has direct visibility, grows by $\varphi^2$. Therefore, the number of view factors grows by $\varphi^4$. The separation distance in terms of the number of grid boxes between each pair of mutually visible faces, which determines the time needed to perform the raytracing for such ray, grows also by $\varphi$ and, therefore, the total raytracing time grows by $\varphi^5$.

In order to analyse the scalability of the algorithm, assume that the number of processes used for the calculation grows by

the same factor as the size of the 3-D grid, i.e. by $\varphi^3$. In this situation, the computational demands of each process grow by $\varphi^2$ and also the proportion of interprocess MPI data exchange relative to the process local memory access increases in both the raytracing and the time-stepping part of RTM, so the process does not scale well.

The situation is better in the first case where the domain of size $n \times n$ grid points grows horizontally, and the average terrain height does not change. For typical terrain and building profiles, the average distance of the visible horizon does not increase

with the horizontal scaling, or it increases much less than linearly. That also means that the average number of other faces, to which each face has direct visibility, does not increase significantly. This property also helps to keep the computation more localized for parallelism. However, these assumptions are valid for a typical scenario only, while the worst case complexity is still in the order of $\mathcal{O}(n)$ for raytracing distance and $\mathcal{O}(n^5)$ for the total computational demands of raytracing.

### 2.2.3 Reducing the number of view factors in the legacy discretization

To reduce the high number of view factors with the legacy discretization scheme, RTM allows to exclude some view factors that are considered less important. First, a minimum value $F_{\min}$ of the irradiance factor can be specified. When faces $i$ and $j$ are mutually visible but the source face $i$ occupies so small portion of the view from the target face $j$ that the value of the irradiance factor $\frac{\tilde{F}_{i \to j} A_i}{A_j} = \tilde{F}_{j \to i}$ is less than $F_{\min}$, then this irradiance factor is disregarded. Thanks to the fact that the potential value of the irradiance factor $\frac{\tilde{F}_{i \to j} A_i}{A_j}$ can be established even before the raytracing from face $i$ to face $j$ begins, the

raytracing is skipped altogether for such face pairs. In addition to the minimum irradiance factor value, a maximum raytracing distance $s_{\max}$ can also be specified. This limit avoids starting the computationally intensive raytracing routine for such face pairs where the mutual distance is above the limit.

The normalization described in Section 2.2.1 ensures that the remaining irradiance factors are increased accordingly in order to maintain the energy conservation by the condition

$$\sum_{m=1}^{n} \frac{\hat{F}_{m \to j} A_m}{A_j} + F_j^{\mathrm{s}} = 1$$





where $\hat{F}_{m \to j}$ is the normalized irradiance factor from face $m$ and $F_j^{\mathrm{s}}$ is the sky view factor representing the view towards sky (described later in Section 2.4). Both $F_{\min}$ and $s_{\max}$ have to be chosen carefully considering the geometry of the modelled domain so that the impact on radiative energy balance in the model is not too high.

### 2.2.4 Angular discretization of the view

The asymptotic complexity of the legacy scheme does not allow simulations of very large domains with horizontal sizes in the order of millions of grid boxes or more. Furthermore, if the view from some face is composed by both very close and very distant faces, the computational resources are used unevenly: proportionally less resources are spent on close faces, each of which represents a higher share of the face's view and potentially also greater share of its irradiance, while more resources, often a majority, are spent on less relevant distant faces. Neglecting the smaller view factors as described in Section 2.2.3 and

normalizing the result is also a possible way to combat this disproportion. This approach, however, has to be used carefully, because it can significantly alter the ratio between face's irradiance from close and from distant surfaces, which could introduce a systematic bias in radiant fluxes coming from the close and distant surfaces.

Thus, we introduce the new angular discretization scheme for reflected and emitted radiation. The general motivation for this approach is based on the observation that the properties of most surfaces are smooth in space and thus two faces next to each

other tend to have similar properties and radiate similarly more often than two generic unrelated faces. This consideration leads to the idea of representing a target face's irradiation from multiple neighbouring distant faces by a single view factor that uses the radiation from one of them but its view factor value represents all of them. This approach allows to use the computational resources more efficiently.

The angular discretization scheme divides the view from each face into a fixed number of directions specified by uniformly

distributed azimuth and elevation angles, as opposed to the uneven set of directions towards the centres of every other visible face in the legacy discretization scheme. Raytracing is performed towards this fixed set of directions with considerable optimization due to the fact that multiple rays of this set share an identical horizontal direction (i.e. azimuth, see Section 3.2). For each ray, the face that covers the first detected obstacle (terrain or building) is used to create a view factor entry. Its view factor value represents exactly the portion of the view corresponding to its direction segment (the section of azimuths and elevations instead of being determined by the other face's size and position). Figure 4 depicts the geometry of the discretization and also

demonstrates the Nusselt analogue, which can be used to visualize the relative sizes of the view factor values.

This approach is equivalent to the raytracing algorithm used in computer graphics, the only difference is that the ray directions in computer graphics correspond to individual pixels of the simulated camera's sensor and often some supersampling is used for anti-aliasing. This similarity demonstrates that the result of this raytracing arrangement represents a reasonable

simplification of view from the selected target, and also that the accuracy can be improved as needed by increasing the angular resolution, i.e. the number of discretized azimuth and elevation angles.

An additional benefit of the angular discretization is the fact that the view factor values, if calculated analytically, always add up exactly to one and there is no need for normalization. A single face often represents an obstacle detected in more than one direction. In such cases, the respective view factors are aggregated to save resources. For faces very close to each other, the

**Geoscientific Model Development** Open Access
Discussions

**Figure 4.** A 3-D representation of the angular discretization scheme for a horizontal face (left) and a vertical face (right). The top row figures depict a view from the centre of a horizontal and a vertical face, where the view has been divided regularly by a fixed number of azimuth and zenith angles, as shown by the half-spheres. The green arrow indicates the traced ray representing the selected angular section, which passes through the centre of that section. The bottom row demonstrates the Nusselt analogue, where the area of each angular section's intersection with the half-sphere, as projected in the orthographic projection to the face's plane (solid red area), is directly proportional to the corresponding view factor value as a portion of the whole view. From the relative sizes of the projected areas it is clear that the view is less uniformly divided for the vertical faces, yet the unified discretization has computational benefits.





sum of the view factor values representing those directions is typically more precise than the normalized approximate value

calculated using (4) just for centres of the grid boxes.

### 2.2.5   View factor values for angular discretization

With angular discretization, the view from the centre of each face is divided into sections, each of which is bounded by

azimuth angles $[\alpha_0, \alpha_1]$ and zenith angles $[\theta_0, \theta_1]$ (see Figure 4). The portion of view represented by such section is calculated

analytically by integration.

The section of view between $[\alpha_0, \alpha_1]$ and $[\theta_0, \theta_1]$ can be viewed as an imaginary surface $A_j$ on a unit sphere. The calculation

of the view factor value is based on the view factor integral (2) where the sending surface $A_i$ is replaced by the centre point

$C_i$, therefore the integral $\frac{1}{A_i} \int_{A_i} \ldots \, \mathrm{d}A_i$, which provides spatial average over the surface $A_i$, is eliminated and only the integral

over $A_j$ remains:

$$F_{C_i \to j} = \int\limits_{A_j} \frac{\cos \beta_{\hat{i}} \cos \beta_{\hat{j}}}{\pi r_{\hat{i}\hat{j}}^2} \, \mathrm{d}A_{\hat{j}} .$$

$A_j$ is a section of a sphere with centre $C_i$ and radius $r = 1$ limited by $[\alpha_0, \alpha_1]$ and $[\theta_0, \theta_1]$. A ray from $C_i$ towards a surface

element $\mathrm{d}A_{\hat{j}}$ is always perpendicular on it, giving $\cos \beta_{\hat{j}}(\alpha, \theta) = 1$. With this and $\mathrm{d}A_{\hat{j}} = r_{\hat{i}\hat{j}}^2 \sin\theta \, \mathrm{d}\alpha \, \mathrm{d}\theta = \sin\theta \, \mathrm{d}\alpha \, \mathrm{d}\theta$ the view

factor value equals

$$F_{[\alpha_0,\alpha_1],[\theta_0,\theta_1]} = \int\limits_{\alpha=\alpha_0}^{\alpha_1} \int\limits_{\theta=\theta_0}^{\theta_1} \frac{\cos \beta_{\hat{i}}(\alpha,\theta) \cos \beta_{\hat{j}}(\alpha,\theta)}{\pi r_{\hat{i}\hat{j}}^2} r_{\hat{i}\hat{j}}^2 \sin\theta \, \mathrm{d}\alpha \, \mathrm{d}\theta = \frac{1}{\pi} \int\limits_{\alpha=\alpha_0}^{\alpha_1} \int\limits_{\theta=\theta_0}^{\theta_1} \cos \beta_{\hat{i}}(\alpha,\theta) \sin\theta \, \mathrm{d}\alpha \, \mathrm{d}\theta . \tag{5}$$

For a horizontal face, the normal angle $\beta_{\hat{i}}(\alpha, \theta)$ is independent of the azimuth angle $\alpha$ and equal to the zenith angle $\theta$:

$$F_{[\alpha_0,\alpha_1],[\theta_0,\theta_1]} = \frac{1}{\pi} \int\limits_{\alpha=\alpha_0}^{\alpha_1} \int\limits_{\theta=\theta_0}^{\theta_1} \cos\theta \sin\theta \, \mathrm{d}\alpha \, \mathrm{d}\theta = \frac{\alpha_1 - \alpha_0}{\pi} \int\limits_{\theta=\theta_0}^{\theta_1} \cos\theta \sin\theta \, \mathrm{d}\theta = \frac{\alpha_1 - \alpha_0}{2\pi} \int\limits_{\theta=\theta_0}^{\theta_1} \sin 2\theta \, \mathrm{d}\theta =$$

$$= \frac{(\alpha_1 - \alpha_0)(\cos 2\theta_0 - \cos 2\theta_1)}{4\pi} . \tag{6}$$

In case of a vertical face, the calculation depends on the orientation of the face. The calculation is presented for a northward

oriented face, for which the face normal $(\alpha_N, \theta_N) = (0, \frac{\pi}{2})$. Considering the spherical triangle formed by the face normal,

zenith and $(\alpha, \theta)$, the central angle $\cos \beta_{\hat{i}}(\alpha, \theta)$ between the face normal and $(\alpha, \theta)$ is calculated using the spherical law of

cosines:

$$\cos \beta_{\hat{i}}(\alpha, \theta) = \cos \theta_N \cos\theta + \sin\theta_N \sin\theta \cos |\alpha_N - \alpha| = \sin\theta \cos\alpha ,$$

and the view factor value is

$$F_{[\alpha_0,\alpha_1],[\theta_0,\theta_1]} = \frac{1}{\pi} \int\limits_{\theta=\theta_0}^{\theta_1} \int\limits_{\alpha=\alpha_0}^{\alpha_1} \cos\alpha \sin^2\theta \, \mathrm{d}\alpha \, \mathrm{d}\theta = \frac{(\sin\alpha_1 - \sin\alpha_0)(\theta_1 - \theta_0 + \sin\theta_0 \cos\theta_0 - \sin\theta_1 \cos\theta_1)}{2\pi} . \tag{7}$$





### 2.2.6 Computational complexity of the angular discretization

The angular discretization scheme greatly improves scalability, which can be demonstrated by following the two scaling approaches introduced in Section 2.2.2. In the case of angular discretization, the number of view factors and the memory requirements are limited by a fixed number for each face and thus the asymptotic order of their growth is $\mathcal{O}(\varphi^2)$ even in the case of
increasing the resolution of the domain (the second case from Section 2.2.2).

The CPU time and inter-process communication demands for raytracing are slightly higher than that, because the average separation distance (i.e. raytracing length) grows with increasing of resolution. For horizontal domain enlargement, only some raytracing directions will have greater distances, while for increasing resolution, all distances will be proportionally longer. In both cases, the demands are in the order of $\mathcal{O}(\varphi^3)$ at worst, which is a great improvement from $\mathcal{O}(\varphi^5)$. Furthermore, we have to
consider that for any atmospheric model, the complexity of increasing resolution in the turbulent flow solver is between $\mathcal{O}(\varphi^3)$ (considering only the increased resolution in three dimensions) and $\mathcal{O}(\varphi^4)$ (also accounting for the shortened time step), so the radiative part can still theoretically scale better than the rest of the model.

### 2.3 Representing semi-transparent plant canopy

The resolved plant canopy in RTM is represented as a 3-D discretized field of leaf area density. RTM simulates the absorption
of SW and LW radiation from the sun, the sky and modelled surfaces (i.e. shading by plants) and the thermal emission of LW radiation from plant canopy towards the sky and the surfaces (see Figure 1). The raytracing algorithm follows the ray from source to target and the attenuation is quantified for each PCGB that the ray intersects. Some other plant canopy related radiative processes are intentionally omitted, namely:

1. Radiative interaction within plant canopy itself by means of LW radiation, that is interaction among individual PCGBs.
This simplification has two reasons. The number of PCGBs can be much higher than the total number of faces in certain scenarios, generating huge amounts of mutually visible pairs of PCGBs and it would be too complex to simulate it with the available computational resources—not to mention the complexity of including the sub-grid part of this interaction. On the other hand, if the LAD is high, then most of LW interaction takes place between neighbouring PCGBs, and because the structure of air temperature is usually very smooth, such PCGBs have low temperature difference, making
the net exchanged radiative flux negligible; if the LAD is low, then its emitted LW flux density is also low.

2. Reflection in both parts of radiation spectrum. The structure and arrangement of plant leaves allows for multiple reflections, but most of these reflections occur between leaves close to each other, which is mostly a sub-grid process (in typical resolutions of units of metres). Moreover, the high emissivity of leaves (and therefore low reflectivity according to Kirchhoff's law) makes their LW reflections negligible.

The main object of radiative modelling in RTM are surfaces; plant canopy is part of the process, but the focus remains on its interaction with surfaces. The data structures are organized accordingly and the raytracing algorithm is adapted to that as well.





This arrangement allowed to add modelling of LW plant canopy emission and absorption into RTM with no data overhead and a negligible increase in computational time.

### 2.3.1 Calculating plant canopy sinks

As the raytracing algorithm follows the ray from source to target, the attenuation is quantified for each PCGB that the ray intersects. Since the raytracing is performed during the initialization phase of the simulation, the actual radiant flux carried by the ray is not yet known, but the attenuation can be expressed as the absorbed fraction of the flux that enters the PCGB. This fraction remains constant in time and independent on the absolute value of the radiant flux, as long as the leaf area density, on which the optical density of the plant canopy is based, remains constant. For this reason the RTM currently does not allow

changing the LAD values during simulation time, which is usually not a problem for typical simulations lasting several days.

The plant canopy within the volume of each PCGB is considered homogeneous. The ratio of the flux $\Phi^{\mathrm{t}}$ passing through the grid box to the flux carried by the ray upon entering the box $\Phi^{\mathrm{i}}$ is called transmittance ($T^{\mathrm{r}}$) and it can be calculated as

$$T^{\mathrm{r}} = \frac{\Phi^{\mathrm{t}}}{\Phi^{\mathrm{i}}} = e^{-\alpha a s},\tag{8}$$

where $a$ is the leaf area density, $s$ is the length of ray's intersection with plant canopy and $\alpha$ is the extinction coefficient, which

converts LAD of trees and shrubs to a corresponding average optical density. The absorbed fraction $\Phi^{\mathrm{a}}$ of the entering flux $\Phi^{\mathrm{i}}$ is then calculated as $\Phi^{\mathrm{a}} = (1 - T^{\mathrm{r}})\Phi^{\mathrm{i}}$. Equation 8 follows and extends the way the absorption of radiative flux in plant canopy is calculated for the single-column case with aggregated *leaf area index* (LAI) in PALM plant canopy module (PCM, see Maronga et al., 2015).

For a ray that passes sequentially through PCGBs $1 \ldots n$ with transmittances $T^{\mathrm{r}}_1, \ldots, T^{\mathrm{r}}_n$, the total transmittance equals to

$T^{\mathrm{r}} = \prod_{m=1}^{n} T^{\mathrm{r}}_m$. The fraction of absorbed flux $\Phi^{\mathrm{a}}_i$ at grid box $i$ divided by the total radiant flux $\Phi^{\mathrm{r}}$ carried by the ray at its origin can be expressed as:

$$F^{\mathrm{rc}}_i = \frac{\Phi^{\mathrm{a}}_i}{\Phi^{\mathrm{r}}} = \frac{\Phi^{\mathrm{a}}_i}{\Phi^{\mathrm{i}}_i} \cdot \frac{\Phi^{\mathrm{i}}_i}{\Phi^{\mathrm{r}}} = (1 - T^{\mathrm{r}}_i) \prod_{m=1}^{i-1} T^{\mathrm{r}}_m = (1 - T^{\mathrm{r}}_i)\left(1 - \sum_{m=1}^{i-1} \frac{\Phi^{\mathrm{a}}_m}{\Phi^{\mathrm{r}}}\right).$$

This fraction, which will be further called the *ray canopy sink factor* (RCSF), is computed iteratively during the raytracing process and it is stored for each intersection of a ray and a PCGB. The total transmittance $T^{\mathrm{r}}$ of the whole ray from source to

target face is stored alongside the respective view factor and the computed irradiance of the target face from that ray is always reduced accordingly.

The total flux $\Phi^{\mathrm{r}}_{j \to k}$ carried by the ray from face $j$ to face $k$ is equal to

$$\Phi^{\mathrm{r}}_{j \to k} = J_j A_j F_{j \to k},$$

where $A_j$ is the area of the source face $j$. The value of the absorbed flux can be obtained by multiplying this value by the

RCSF. The flux absorbed at PCGB does not depend on the ray's target and only the total absorbed flux for each PCGB needs to be calculated:

$$\Phi^{\mathrm{a}}_{i,j} = \sum_m \Phi^{\mathrm{a}}_{i,j \to m} = \sum_m \frac{\Phi^{\mathrm{a}}_{i,j \to m}}{\Phi^{\mathrm{r}}_{j \to m}} \Phi^{\mathrm{r}}_{j \to m} = \sum_m F^{\mathrm{rc}}_{i,j \to m} J_j A_j F_{j \to m}.$$





Thanks to that, all RCSFs with the same source face and PCGB (i.e. those that differ only by the target face) can be aggregated. They are multiplied by the appropriate view factors and the resulting sum $F_{i,j}^{\mathrm{c}}$ is called the *canopy view factor* (CVF):

$$F_{i,j}^{\mathrm{c}} = \frac{\Phi_{i,j}^{\mathrm{a}}}{J_j A_j} = \sum_m F_{i,j\to m}^{\mathrm{rc}} F_{j\to m} \, . \tag{9}$$

This aggregation reduces storage and computation demands during the post-initialization time stepping part of the simulation. The CVF only needs to be multiplied by the area of the source face and source face's radiosity in order to obtain the radiant flux $\Phi_{i,j}^{\mathrm{a}}$ absorbed by PCGB $i$ originating from face $j$.

### 2.3.2 Thermal emission from plant canopy

Modelling of the plant canopy thermal emission follows the concept outlined earlier. The emission from the plant canopy is considered from the target face's point of view while the internal LW radiation exchange inside plant canopy (among individual PCGBs as well as the intra-grid exchange) is omitted.

Due to the reciprocity property of view factors, the CVF actually represents the fraction of view from face $j$ that is covered by plant canopy from PCGB $i$, taking into account the partial opacity, which is determined by the leaf area density in PCGB $i$.
Consider the sub-grid semi-transparency to be caused by randomly distributed, small, fully opaque leaves with fully transparent gaps in between them, then the CVF is exactly equal to the view factor from face $j$ towards those leaves (i.e. their visible parts).

This enables straightforward modelling of the thermal emission originating from the leaves in PCGB $i$ that is absorbed by the face $j$. Since the reflection in plant canopy is ignored, the emissivity of those leaves can be considered 1 according to the Kirchhoff's law, and using the Stefan–Boltzmann law, the emitted radiative flux from PCGB $i$ in the direction of the face $j$ is
equal to

$$E_{i\to j} = F_{i,j}^{\mathrm{c}} \sigma T_i^4 \, , \tag{10}$$

where $T_i$ is the temperature of the air and leaves inside the PCGB $i$.

Thermal emission from plant canopy towards the sky has to geometrically match the absorbed LW radiative flux from the sky in order to avoid biases in total energy budget of the modelled domain. It is computed in a similar manner, using special
CVF entries which have the sky as the source instead of a face (their calculation is described later in Section 3.2).

### 2.4 Discretization of the direct and diffuse solar radiation

The direct and diffuse components of the incoming solar radiation and the thermal radiation from the sky towards surfaces are represented using the *sky view factor* (SVF). It represents the portion of view from individual faces towards the sky which is not occupied by other faces. If the sky is viewed as an imaginary face, SVF makes the system of faces enclosed as specified in
(1) which can be expressed as

$$F_i^{\mathrm{s}} = 1 - \sum_{j=1}^{n} F_{i\to j} \, .$$





The radiant fluxes from the sky propagate through the urban layer similarly to the reflected an emitted radiation from the faces with the exception that the source lies outside the urban layer. As the intention of the design is to avoid raytracing during model time-stepping for the reasons explained in Section 2.1.1, all raytracing representing these rays is done also in advance during the initialization phase of the model just like with the other rays.

RTM version 3.0 represents the sky by a single SVF. The value of $F^s$ is calculated by 2-D raytracing as described in Section 3.2. For homogeneous diffuse solar radiation, this single value per face is sufficient to allow calculation of the diffuse solar irradiance. It is also possible to consider diffuse solar radiation as inhomogeneous by splitting the sky to multiple regions and storing such separate partial SVF values per face because the number of regions would not increase with domain size; the current RTM code is ready for addition of this option once such radiation data is available.

When plant canopy simulation is disabled, the only information necessary to calculate the SVF for a specific location are the horizon heights in each discretized azimuth direction as is described in detail in Section 3.2. When plant canopy reduces transmissivity from the sky, the vertical structure of this partial shading is calculated in the means of angular discretization by adding a fixed number of discretized elevation angles for which the transmissivity of the path from the sky towards the target face is calculated.

### 2.4.1 Direct solar radiation

The calculation of direct solar radiation during the model initialization phase is complicated by the fact that the apparent position of its source, the sun, and therefore the geometry of all rays, changes throughout the day, while all the other radiation sources in the model have fixed positions and geometries and only the values of their radiant fluxes change in time. RTM solves this problem by discretization of the apparent solar positions and performing raytracing between these predetermined apparent solar positions and corresponding faces during the model initialization.

For a typical simulation which spans times in the order of hours or days, there is a fixed number of apparent solar positions (at most the number of radiation time steps), which is further reduced by discretization of azimuth and elevation angles and using the nearest discretized direction. RTM uses the discretized directions that are already used for calculation of $F^s$ in order to optimize the computation as much as possible.

For each the discretized direction $D_j$ and face $i$, the total ray transmittance $T^{\mathrm{r}}_{D_j \to i}$ is stored. By multiplying it by the solar normal irradiance $E^d$ and by the cosine of the incident angle, the approximate direct irradiance $\tilde{E}^d_i$ of face $i$ is obtained:

$$\tilde{E}^d_i = E^d T^{\mathrm{r}}_{D_j \to i} \cos \theta_i \,,$$

where $E^d$ is the current solar direct normal irradiance and $\theta_{D_j, i}$ is the angle between the normal to face $i$ and the current apparent solar position.

### 2.4.2 Direct irradiation of plant canopy

As described in Section 2.3.1, the canopy view factors represent the partial absorption of radiation by plant canopy only for the radiation that originates from surfaces and for the diffuse solar radiation and thermal emission from the sky directed towards





the surfaces. The absorbed direct solar radiation, which accounts for the majority of absorbed radiative flux during clear-sky
days, needs to be modelled separately.

In order to determine the direct radiative flux entering each PCGB with respect to shading by obstacles and partial shading
by other PCGBs, RTM performs separate raytracing procedure starting backwards from the centre of each PCGB towards
the discretized apparent solar directions. For this raytracing process, no canopy view factors are stored and only the total ray
transmittance is determined and stored for each PCGB $k$ and discretized direction $D_j$.

During time-stepping, the transmittance of the corresponding ray $T^{\mathrm{r}}_{D_j \to k}$ is multiplied by the direct normal solar irradiance
$E^{\mathrm{d}}$, which provides the radiative flux density entering PCGB $k$. The fraction of this flux which is absorbed by PCGB $k$ is
dependent on the dimensions of the PCGB, on the direction of irradiation and on the LAD of PCGB $k$.

RTM uses a sub-grid discretization model which sends an array of $60 \times 60$ parallel rays (organised to fill a bounding rectangle
that contains the projection of the grid box) and calculates the transmittance of each of the rays using (8). These transmittances
together produce an approximation of the fraction of absorbed flux divided by the flux density of direct normal irradiance.
This fraction is calculated at the beginning of each timestep using known apparent solar position. Thanks to the fact that the
grid is regular and the solar rays are parallel, this fraction is applicable to all PCGBs with a specified LAD. This simulation
is performed with a single LAD value $a_{\mathrm{r}} = 0.9 \max\{a_m\}$ for all PCGBs $m$, and the result is linearized for all PCGBs using a
factor $\frac{a_m}{a_{\mathrm{r}}}$.

**2.4.3   Calculation of mean radiant temperature**

*Mean radiant temperature* (MRT) at a certain point in space is defined as the temperature of an imaginary object for which
that object would be in radiative equilibrium with it surroundings, which means that the absorbed irradiance would be equal
to the emitted radiant exitance. Calculation of the MRT is closely related to the radiative processes in the RTM and thus it
is implemented with advantage inside this module. This allows to use a similar approach and reuse existing routines; it also
ensures that MRT is calculated with the same discretization scheme as the scheme used in RTM for the calculation of LW and
SW radiation which allows to avoid utilization of some highly simplified, yet common approaches. Calculated MRT values are
available directly in RTM in the form of PALM output variables as well as they are provided to the biometeorology module for
calculation of biometeorological quantities related to human thermal comfort (see Section 4.4).

Considering both LW and SW radiant fluxes for a hypothetical object with emissivity $\varepsilon$ and SW albedo $a$, the MRT value
$T_{\mathrm{R}}$ can be derived from its defining equivalence and from the Stefan–Boltzmann law:

$$(1-a)E_S + \varepsilon E_L = \varepsilon \sigma T_{\mathrm{R}}^4,$$

where $E_S$ is the average SW irradiance of the hypothetical object and $E_L$ is its average LW irradiance.

The calculation of the MRT utilises similar concept as the calculation of irradiance with angular discretization. For each
point where MRT would be simulated, the *MRT factors* are calculated during the initialization phase of the model run. MRT
factors are the equivalent of irradiance factors for average surface of the hypothetical sphere, that is without dependence on





direction of irradiance. For face $i$ and MRT point $j$, the MRT factor is calculated as

$$\frac{\tilde{F}^{\mathrm{M}}_{i \to j}}{A_j} = \frac{\cos \theta_{C_i}}{4\pi s^2_{C_i,j}}, \tag{11}$$

where $A_j$ is surface area of the sphere, $\theta_{C_i}$ is the angle between the connecting ray and the normal of face $i$ and $s_{C_i,j}$ is the length of the connecting ray. The equation (11) utilizes the fact that the ratio between surface of a sphere and its normal projected area (the area of a circle with the same diameter) is equal to 4.

The MRT factors are precalculated using the 2-D raytracing algorithm with angular discretization of the whole view, together with MRT sky-view factors and direct solar irradiance transmissivities for each MRT point. Depending on configuration, the MRT can be calculated for the centre of every grid box in the first layer above terrain or even in multiple vertical layers.

The pure physical MRT value is usually defined with respect to a spherical black-globe thermometer. On the other hand, 480 the biometeorology applications require the MRT value related to a clothed human body, which is tall and narrow and it is therefore affected by radiation from its sides proportionally more than by radiation directly from above. It is modelled in RTM as a configurable asymmetrical generic object with specified albedo, emissivity and aspect ratio. These MRT values for the hypothetical human body are then provided to the biometeorology module as descibed in Section 4.4.

## 3 Implementation of RTM

### 3.1 Basic raytracing algorithm

The initialization step which includes raytracing between faces and establishing view and other required factors is organized in a way which optimizes their actual usage during the simulation time step phase. As the view factor $\hat{F}_{i \to j}$ is used to determine the irradiance of face $j$, the view factors are computed in the MPI process which controls the subdomain where their *target* faces lie, so that the results of the raytracing do not need to be transferred among processes.

The raytracing process is called within a nested loop. The outer loop iterates over all target faces belonging to local process's subdomain. The inner loop iterates all faces from the whole modelled domain, treating them as sources for raytracing. During the initialization step, an index of all faces and their coordinates is generated and distributed among processes using the MPI `gather` operation. A typical urban scenario has the number of faces proportional to horizontal size of the domain and an index of all faces can fit in each process's memory.

The first test in the inner loop skips any source faces which are not oriented towards the processed target face or this target face is not oriented towards them (see Section 2.1.1). For each remaining source face, the potential view factor value is calculated using (4); this value is valid if the raytracing does not discover any opaque obstacles between the two faces. This value needs to be known in advance because it is used when creating CVF entries (see (9)). In case of legacy discretization with reducing of the number of view factors (Section 2.2.3), the raytracing is skipped for the face pair if the potential view 500 factor value is less than specified minimum value or if the separation distance is above the specified maximum. Otherwise, the actual raytracing is performed, starting from the source face towards the target face.





Tracing the ray which travels through a regular grid means generating a sorted list of grid boxes that the ray intersects. The raytracing algorithm takes advantage of the regularity, orthogonality and equidistance of the grid. A ray that travels from coordinates $[x_1, y_1, z_1]$ to coordinates $[x_2, y_2, z_2]$ necessarily crosses exactly those dividing planes in dimensions $yz$ that are located between $x_1 \ldots x_2$, the $xz$ planes that are located between $y_1 \ldots y_2$ and the $xy$ planes that are located between $z_1 \ldots z_2$, only the order in which these crossings are intermixed is not yet known.

Due to computational efficiency, while tracing the ray the RTM uses a coordinate system in which the planes separating grid boxes have integer coordinates. Therefore each grid box centre has coordinates $[x + \frac{1}{2}, y + \frac{1}{2}, z + \frac{1}{2}]$ for some $x, y, z \in \mathbb{Z}$. It means that in case of $x_1 < x_2$, the list of crossed $yz$-plane boundaries is $\lceil x_1 \rceil \ldots \lfloor x_2 \rfloor$, and in case of $x_1 > x_2$, the list is $\lceil x_2 \rceil \ldots \lfloor x_1 \rfloor$, and similarly for other dimensions.

The raytracing cycle follows this general pattern:

1. Set current position to the starting point.

2. Determine the Euclidean distance in raytracing coordinates between the starting point and the next separating plane which immediately follows the current position. This is done for each dimension $x$, $y$ and $z$ separately,

3. Select the dimension with the smallest distance to the next separating plane, mark this as the next grid crossing.

4. Identify the grid box between the current position and the next grid crossing.

5. Determine whether that grid box lies above or below the terrain in 2.5-D geometry. In the latter case, the ray is blocked by an opaque obstacle and the raytracing cycle ends prematurely with negative result for the face pair.

6. Perform all tasks scheduled for each visited grid box.

7. Advance the current position to the next grid crossing. For the selected dimension, advance one separating plane further, this will become the next separating plane for the selected dimension.

8. Repeat from step 3 until reaching the target point.

Because the ray might cross multiple separating planes at once and in order to avoid numerical instability, steps 4–6 are skipped if the length of the ray's intersection with the grid box is less than 0.001 in raytracing coordinates. The only limitation of this algorithm is that the whole ray must not lie within one separating plane, but it can be easily demonstrated that this cannot happen when the raytracing is performed between centres of two faces that are oriented towards each other, nor may it happen for rays going to and from grid box centres. The map of terrain elevations for the whole domain, which is needed in the step 5, is represented by a 2-D array. It does fit in the memory of each MPI process and it is prepared in advance using the MPI `gather` operation, therefore the obstacle detection is a fast operation.

## 3.2 Azimuthal raytracing algorithm

With the introduction of the angular discretization in RTM version 2.0, a new variant of the raytracing algorithm was developed which was highly optimized for this angular discretization. This algorithm is further called *2-D raytracing*. The algorithm





utilizes following feature of the 2.5-D geometry: for every point of view and for every azimuth exists a distinct *horizon* ($\gamma$), i.e. the elevation angle below which the view is completely obstructed by terrain and/or buildings and above which there is only

sky. The extension of this alghoritm to a full 3-D geometry is discussed in Section 6.

      The core of the 2-D raytracing algorithm works by following a discrete set of azimuths from point $(x, y, z)$ (representing centre of the target face) in direction of the azimuth until it reaches horizontal domain border. For each azimuth it tracks the monotonically increasing horizon angle; more specifically, it tracks $\tan\gamma = \frac{z_h - z}{\sqrt{(x_h - x)^2 + (y_h - y)^2}}$ where $(x_h, y_h, z_h)$ are coordinates of the obstacle representing the tracked highest horizon angle. The tracking itself works just like the basic raytracing

algorithm (Section 3.1) except in just two dimensions—one step means one vertical column for which the terrain and building height is compared against the currently known horizon.

      To determine the partial shading by plant canopy, RTM needs to track more than just the horizon angle for each traced azimuth. The plant canopy may have diverse vertical structure, thus an evenly discretized set of elevation (zenith) angles is tracked for each azimuth. This forms a uniform, regular set of directions, which is used for all types of the radiative processes;

it is used for calculation of the sky view factors, direct irradiance transmissivity and also for the angularly discretized view factors towards other surfaces. This way, a single 2-D raytracing routine computes all the respective values at once without any overhead.

      During tracing of each ray, the information about LAD along all the ray path is needed. This information is distributed in particular MPI processes and needs to be obtained by means of MPI communication. In order to reduce fragmentation of one-

sided MPI operations, the 2-D raytracing requests all LAD data for all applicable PCGBs belonging to the whole half-plane cross section (one discrete azimuth) in all required vertical levels at once. When these data are retrieved from all involved MPI precesses, the RCSFs are generated in a two-pass calculation for each discrete azimuth—from point $(x, y, z)$ towards the horizon and then back. Both of these directions are necessary, one for the absorbed fraction of incoming diffuse radiation from the sky, the other one for the absorbed fraction of the outgoing (reflected and emitted) radiative flux towards the sky. In addition

to that, the total transmittance of each ray is saved. The generated RCSFs are sorted and aggregated continuously and, after all raytracing is done, redistributed among processes using MPI `alltoallv` similarly as with the basic raytracing algorithm.

      The 2-D raytracing algorithm needs to determine the complete information for the timestepping radiation calculation: the index of the opposing face, the view factor value and the total transmissivity of the connecting ray, for each discrete direction under the horizon angle. As is specified in Section 2.2.4, the view factor value is determined solely by the regularly discretized

fraction of view. For vertical faces, it is calculated using (7) where $\alpha_0$ and $\alpha_1$ are the midpoints between the calculated discrete azimuth and the neighbouring discrete azimuths (or $-\frac{\pi}{2}$ or $\frac{\pi}{2}$, where the calculated discrete zenith angle is the first or the last value, respectively) and $\theta_0$ and $\theta_1$ are the midpoints between the calculated discrete zenith angle and the neighbouring discrete zenith angles (or $0$ or $\pi$ as boundaries). For horizontal faces, similar approach is used with (6).

      The index of the opposing face has to be determined using MPI one-sided communication request (MPI `get`), because the

array with reverse indices $(x_i, y_i, z_i, d_i) \rightarrow i$ (where $d_i$ is the face orientation—northward, eastward, southward, westward or upward) is once again a 3-D array for which each process can only hold its own subdomain in its local memory, as the array for the whole domain would be too large.



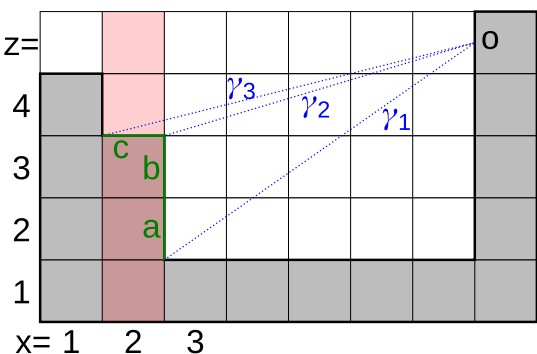

**Figure 5.** Obstacle identification algorithm (vertical cross-section).

For each new grid column processed during raytracing, there may be at most one new horizontal face and zero or more vertical faces identified as new opposing faces. The identification algorithm can be demonstrated on an example shown in

the figure 5. The raytracing procedure, originating from face $o$ which progresses in the azimuth direction corresponding to the cross-section in the figure, enters column $x = 2$ with the current horizon angle $\gamma_1$, which was the result of column $x = 3$ having terrain height $z = 1.5$. The column $x = 2$ has terrain height $z = 3.5$, which yields two new horizon angles $\gamma_2$ and $\gamma_3$ for entry to the column ($x = 2.5$) and exit from the column ($x = 1.5$) respectively. Because $\gamma_2 > \gamma_1$, there will be new vertical opposing faces for each discretized ray between $\gamma_1 \ldots \gamma_2$, in this case faces $a$ and $b$. The orientation of these faces is determined by the

fact that the boundary between columns was in dimension $x$ decreasing, i.e. they are eastward oriented faces. Furthermore, because $\gamma_3 > \gamma_2$, then as long as there is at least one discretized ray between $\gamma_2 \ldots \gamma_3$, there will also be a new opposing horizontal face $c$.

The generated VF entries for the opposing faces are sorted and aggregated for each raytracing origin (after raytracing towards all discretized azimuth angles), creating at most a fixed number of entries that do not need to be normalized, as described in

Section 2.2.4. VF entries are always generated in the process computing the subdomain where the target face lies, therefore there is no need for their redistribution.

### 3.3 Radiation processing in time-stepping

RTM radiation interaction is called in PALM after every call of the one column radiation scheme (e.g. RRTMG), which is applied in regular configurable intervals. For each radiation step, the radiative fluxes on the top of the urban canopy layer are

updated first, and then the RTM calculates the fluxes within the urban layer using inputs from its top border.

The calculation of radiative fluxes by RTM within time-stepping is much less computationally demanding than calculations performed during the initialization phase. Memory usage is slightly lower than during initialization, because the VF and CVF entries have been fully aggregated and stored in size-optimized arrays.





The RTM time step begins with calculation of values that depend solely on time, like solar geometry and derived values (e.g.
precalculated absorptivity values for a prototype semitransparent box). If the forcing radiative scheme did not provide separate
direct and diffuse horizontal irradiances, the global horizontal irradiance is split into these components using the statistically
derived Boland model (Boland et al., 2008).

### 3.3.1   First atmospheric pass

The next step is updating radiosities for primary sources of radiation. For surfaces, the LW radiant exitance is calculated using
surface temperature and emissivity according to the Stefan–Boltzmann law:

$$M_i = \varepsilon_i \sigma T_i^4,$$

where $\varepsilon_i$ is the surface emissivity and $T_i$ is the surface temperature. The assigned radiosity values are exchanged among
processes using MPI `gather` operation. After that, the first propagation of radiation throughout the domain from sources
to targets (before the first reflection) is simulated. The LW emitted radiant flux is multiplied by irradiance factor values to
calculate the irradiance of target faces.

The VF entry cycle is followed by straightforward application of diffuse irradiance with the SVF. The direct solar irradiance
is applied using lookup table from apparent solar position angles to precalculated direct solar transmittance entries. The value
of direct solar irradiance for face $i$ is calculated as

$$E_i^{\mathrm{d}} = E^{\mathrm{d}} \frac{\cos \gamma_i}{\cos \theta_{\mathrm{s}}},$$

where $E^{\mathrm{d}}$ is the current value of horizontal direct solar irradiance, $\theta_{\mathrm{s}}$ is the solar zenith angle and $\gamma_i$ is the angle between
apparent solar position and the normal of face $i$. The fraction is precalculated for the five possible face orientations at the
beginning of the radiative time step.

When the MRT calculation is enabled, the direct, diffuse and thermal MRT components are calculated using equivalent
entries—MRT factors, SVFs and direct solar transmittances.

The last cycle in the first atmospheric pass before reflections iterates the CVF entries (if plant canopy is present). There are
two types of CVF entries: those representing rays from the sky (one for each PCGB) and those representing rays from surfaces
(per PCGB and source face). For the CVF entries from the sky, the respective fraction of current diffuse irradiance is absorbed
and the fraction of direct solar irradiance is added using lookup table for solar angles.

The CVF entries from faces are used in this step only when plant canopy LW interaction is enabled. In that case, the
respective fraction of transmitted thermal radiant flux is absorbed and at the same time, the emitted radiant flux from the PCGB
towards the respective face is calculated according to (10), stored and subtracted from the absorbed radiant flux. In case the
PCGB and the respective face have identical temperatures, these two fluxes are equal and the net absorbed flux is zero as
expected. The sums of radiant fluxes emitted from plant canopy toward each face are then exchanged among processes using
MPI `gather` and added to the rest of the incident LW irradiance.





### 3.3.2 Iterative reflections

If reflections are enabled, then the remaining steps are repeated $n$-times according to configuration. Each iteration starts with the absorption of the respective fraction (one minus albedo for SW, emissivity for LW) of irradiance from the previous step for each face. The remaining irradiance is reflected—assigned as outgoing radiosity for the current step and exchanged among processes using MPI `gather`. The outgoing radiosity is then propagated once more through the domain using view factors, again reduced by partial shading according to plant canopy configuration.

The next step is calculation of radiative flux absorbed by plant canopy using CVF entries. In the reflected passes, only the entries for flux originating from surfaces are processed. At the end of each reflective iteration, the MRT factors are processed and the reflected radiation is added to the radiative background for MRT points.

After $n$ reflective passes, the remaining incident radiant flux is added completely to the absorbed flux of the respective surface and final calculations are done for the timestep, including converting SW and LW radiances to MRT temperature in Kelvins, converting absorbed radiant flux in plant canopy to volumetric heating rates and calculating plant canopy evapotranspiration rates (see Section 4.3).

## 4 Integration of RTM with other PALM modules

### 4.1 Radiation forcing model

As described in Section 1.2, the RTM simulates radiative fluxes inside the urban canopy layer taking the radiation fluxes from the PALM one-column radiation model as input. As the result of this simulation, RTM calculates the radiation reflected from the surface to the atmosphere more realistically than the one-column model. In order to take advantage of that, the RTM results need to be consider back in the forcing radiation model. This forms a two-way coupling between the forcing radiation model and RTM. This section describes the implementation of the second backward direction of this coupling.

The implementation is based on calculating effective radiation surface parameters for the radiation model: an effective surface emissivity $\epsilon_{\text{eff}}$, surface temperature $T_{\text{eff}}$ and urban albedo $\alpha_{\text{eff}}$. These parameters are calculated so that they would, when applied to a simple single surface as assumed in the forcing one-column radiation model, give similar radiation fluxes as the complex 3-D urban area.

For LW radiation, the lower boundary condition of the forcing radiation model can be expressed as:

$$L^{\uparrow} = \epsilon_{\text{eff}}\sigma T_{\text{eff}}^4 + (1 - \epsilon_{\text{eff}})L^{\downarrow}, \tag{12}$$

where $L^{\uparrow}$ is the upwelling LW radiation, which represents the total radiation emitted and reflected into the sky from the urban surfaces as calculated by the RTM. The downwelling LW radiation $L^{\downarrow}$ is provided by the forcing radiation model as an input to the RTM.

Here, $\epsilon_{\text{eff}}$ is selected as the average of the surface emissivities $\epsilon_i$ of the surface $i$ over the area $A_i$:

$$\epsilon_{\text{eff}} = \frac{1}{A}\sum_i A_i\epsilon_i \quad \text{with} \quad A = \sum_i A_i. \tag{}$$





With that, only $L^\uparrow$ is needed to calculate $T_{\mathrm{eff}}$ with (12). The straightforward way would be to sum up the emitted and reflected radiation from each surface, taking into account the corresponding sky-view factor. For efficiency reasons, the energy conservation for the total urban area is used instead:

$$L^\downarrow + L^{\mathrm{emit}} = L^{\mathrm{abs}} + L^\uparrow.$$

The terms on the left-hand side represents the total energy input from the sky and the total LW emission of the urban surfaces. The right-hand side stands for the total absorbed energy by the urban surfaces as well as the total energy emitted and reflected to the sky. This can be combined with (12) yielding:

$$\epsilon_{\mathrm{eff}}\sigma T_{\mathrm{eff}}^4 = \epsilon_{\mathrm{eff}}L^\downarrow + L^{\mathrm{emit}} - L^{\mathrm{abs}},$$

where

$$L^\downarrow = \frac{1}{A_{\mathrm{norm}}}\sum_i A_i L_i^\downarrow,$$

$$L^{\mathrm{emit}} = \frac{1}{A_{\mathrm{norm}}}\sum_i A_i \epsilon_i \sigma T_i^4,$$

$$L^{\mathrm{abs}} = \frac{1}{A_{\mathrm{norm}}}\sum_i A_i \epsilon_i L_i.$$

Here, $L_i^\downarrow$ is the radiation received by a surface with a temperature $T_i$ from the sky and $L_i$ is the respective total received LW radiation including reflections and LW emission from other surfaces.

The standard choice that the normalizing area $A_{\mathrm{norm}}$ represents the horizontal modelling domain size $A_{\mathrm{horiz}} = l_x \cdot l_y$ with domain size $l_x$ and $l_y$ in $x$- and $y$-direction, respectively, does not work here. In order to receive a realistic amount of diffuse radiation from the sky, it is necessary to consider not only radiation from the sky area of size $A_{\mathrm{horiz}}$ above the modelling domain but also radiation from the side areas of the domain. In general however, this leads to $\sum_i A_i L_i^\downarrow \neq A_{\mathrm{horiz}} \cdot L^\downarrow$, which is not unrealistic because higher (lower) received radiation within the domain would be compensated by lower (higher) received 670 radiation outside of the domain. To tackle this issue, the approach is to receive $L^\downarrow$ from the sky but with a different reference area $A_{\mathrm{norm}}$ calculated as

$$A_{\mathrm{norm}} = A_{\mathrm{horiz}}\frac{\sum_i A_i L_i^\downarrow}{A_{\mathrm{horiz}}L^\downarrow} = \frac{\sum_i A_i L_i^\downarrow}{L^\downarrow}.$$

For SW radiation, the lower boundary condition of the forcing radiation model can be expressed as

$$K^\uparrow = \alpha_{\mathrm{eff}}K^\downarrow$$

with the downwelling SW radiation $K^\downarrow$ as calculated by the the forcing radiation model and the total upwelling SW radiation $K^\uparrow$. Expressing $K^\uparrow$ in terms of absorbed SW radiation $K^{\mathrm{abs}}$ with

$$K^\downarrow = K^\uparrow + K^{\mathrm{abs}}$$





yields

$$\alpha_{\text{eff}} = \frac{K^{\downarrow} - K^{\text{abs}}}{K^{\downarrow}},$$

where

$$K^{\downarrow} = \frac{1}{A_{\text{norm}}} \sum_i A_i K_i^{\downarrow},$$

$$K^{\text{abs}} = \frac{1}{A_{\text{norm}}} \sum_i A_i (1 - \alpha_i) K_i.$$

Here, $K_i^{\downarrow}$ is the SW radiation from the sky received by surface $i$ with albedo $\alpha_i$ and $K_i$ is the total SW radiation received by the respective area including reflections from other surfaces.

## 685  4.2  Building and land surface models

Radiative transfer between atmosphere and surfaces as well as among surfaces themselves depends on the surface temperature, which is the result of the surface energy balance calculated in the surface modules. However, one of the components in the surface energy balance is the surface net radiation, which is calculated in the RTM. The exchange of information between the surface modules and the RTM is therefore mutual.

PALM includes two surface modules: The land surface module (LSM), for natural-like surfaces such as vegetation-covered, water and pavement surfaces, and the building surface module (BSM) for building surfaces such as walls, windows, and roofs. Both modules solve the energy balance for each surface, partitioning the available net radiation into ground/wall heat fluxes, as well as sensible and latent heat fluxes. For a detailed description of LSM and BSM see Maronga et al. (2020).

Each of the discrete surfaces may have distinct soil or wall material properties, such as heat capacity or conductivity, as

well as distinct surface properties such as albedo, thermal emissivity, and roughness length, etc. In the LSM a face (i.e. surface element in LSM and BSM terminology) is assumed to be either vegetation, water, or pavement, while in the BSM a surface element is further divided fractionally into wall, window, and green surface fractions. Each fraction exhibits distinct radiative properties. For performance optimization reasons, the corresponding properties and state variables for the surfaces are stored within a dynamic data structure, which encompasses arrays for various surface variables. Each type of surface with different

spherical orientation has its own derived data structure defined, e.g. north- and southward-facing BSM surfaces can be access individually without further if-else conditions necessary. This way of surface representation allows to execute surface energy related code in a consecutive manner without hampering loop vectorization. However, RTM solves interactions between all surfaces and it thus needs, again for optimization reasons, one single array of surface properties and state variables. Hence, surface information from the respective arrays of the derived data structure are gathered into a single linear array, before the

RTM code is executed. This is done for the surface temperature, albedo and emissivity. For fractional surfaces, these values are calculated as the weighted average of the different fractions (wall, window and green fractions).



After the radiation interactions are performed in the RTM, the resulting LW and SW radiation fluxes at the surfaces are distributed back onto the surface-type data structure. Subsequently, the updated radiation fluxes at the surfaces are supplied to LSM and BSM.

## 4.3 Evapotranspiration and latent heat in plant canopy model

An important process associated with plant canopies is transpiration of water vapour from the green parts of plants. It is actively controlled by plants by opening and closing stomata and thus changing the resistance of the leaf surface against the evaporation of the leaf water. This process is mainly affected by the incoming SW radiation, air temperature, air humidity and by the soil water content (e.g. Stewart, 1988; Daudet et al., 1999). Simulation of detailed structure of SW radiation in RTM allows to create the transpiration model for resolved plant canopy, which calculates humidity gradients and latent heat fluxes, completing the description of the atmospheric thermal energy processes in plant canopy.

Calculation of the plant canopy transpiration rate is based on the Jarvis–Stewart model in the form described in Daudet et al. (1999) and Ngao et al. (2017). Namely, the evaporation rate from the leaf surface $E_{\mathrm{r}}$ is computed as

$$E_{\mathrm{r}} = \Omega E_{\mathrm{eq}} + (1 - \Omega) L E_{\mathrm{imp}},$$

where $E_{\mathrm{eq}}$ is the equilibrium evaporation per leaf unit area, $E_{\mathrm{imp}}$ is the imposed evaporation per leaf unit area, and $\Omega$ is the decoupling factor. These variables are modeled as

$$l_{\mathrm{v}} E_{\mathrm{eq}} = \frac{R_{\mathrm{n}} \frac{q_{\mathrm{s}}}{\gamma}}{\frac{q_{\mathrm{s}}}{\gamma} + 2},$$

$$l_{\mathrm{v}} E_{\mathrm{imp}} = \rho c_p g_{\mathrm{s}} e_{p,d},$$

$$\gamma \Omega = \frac{\frac{q_{\mathrm{s}}}{\gamma} + 2}{\frac{q_{\mathrm{s}}}{\gamma} + 2 + 2 g_{\mathrm{b}}/g_{\mathrm{s}}},$$

where $R_{\mathrm{n}}$ is the net radiation provided by the RTM for each PCGB, $e_{p,d} = e_s - e$ is the water vapour pressure deficit in the air (with $e_s$ and $e$ being the water vapour pressure at saturation and the water vapor pressure, respectively), $q_{\mathrm{s}} = \frac{\partial e_s}{\partial T}$ is the partial derivative of the water vapour saturation pressure with respect to temperature, $\gamma = (c_p p)/(0.622 l_{\mathrm{v}})$ is the psychrometric constant, $g_{\mathrm{b}}$ is the leaf boundary layer conductance, and $g_{\mathrm{s}}$ is the stomatal conductance. The stomatal conductance is computed as

$$g_{\mathrm{s}} = g_{\mathrm{s,max}} f_1(K^{\downarrow}) f_2(T) f_3(e_{p,d}) f_4(\mathrm{RSWC}),$$

where $g_{\mathrm{s,max}}$ is an empirical maximum conductivity value and $f_1 \dots f_4$ are empirical functions, which depend on the incident SW radiation, the temperature, the water pressure deficit and the residual soil water content RSWC (Van Wijk et al., 2000). The empirical functions are adapted from Stewart (1988) and Van Wijk et al. (2000).

The resulting latent heat fluxes and humidity gradients then enter the prognostic equations of humidity and potential temperature (see Eq. 3 and 4 in Maronga et al., 2020) as additional sources terms.





## 4.4 Biometeorology module

The *biometeorology module* in PALM (BIO, see Fröhlich and Matzarakis, 2019) provides spatial and temporal information on the human thermal comfort. This is expressed in form of biometeorology indices, such as physiologically equivalent temperature (PET), universal thermal climate index (UTCI), and perceived temperature (PT). All these indices require the mean radiant temperature (MRT), with respect to a simulated human body.

The calculation of MRT is closely related to the RTM radiative processes. This fact allows to calculate MRT inside RTM with a little additional effort and overhead utilizing the existing RTM routines (see Section 2.4.3). It also ensures that MRT is simulated similarly to other radiative fluxes (i.e. using the same discretization and numerical methods), which allows to avoid substantial simplifications often used in other models (a review is e.g. in Fröhlich and Matzarakis, 2019). However, this approach requires the interconnection and collaboration of the RTM and BIO modules.

The RTM provides the MRT values for the BIO module in the form of separate SW and LW mean irradiance for each simulated MRT box. This approach allows the BIO module to process the incoming fluxes independently and to apply the radiative properties of human body (albedo and emissivity) inside the BIO module. The shape of the simulated body, however, affects the MRT factors and thus it needs to be defined inside the RTM. Current version of RTM contains three selectable types of MRT body geometries: sphere (simulated globe thermometer), ellipsoid, and a simple human body parameterization, with the possibility to supplement other arbitrary geometries. The ratio of the major and minor axes of the elongated shapes is configurable in RTM with the default of 7.3.

## 5 Model evaluation

This section presents evaluation of convergence and computational performance of the current RTM implementation. A validation of the whole PALM model with RTM in a realistic urban environment against a comprehensive set of observations for a large scenario of Prague–Dejvice is presented by Resler et al. (2020). There are also further validation studies (e.g. Berlin) in preparation. A detailed study on the relative importance of individual radiative transfer processes is presented in Salim et al. (2020). Tests of sensitivity of the PALM model to specific RTM input parameters are included in Belda et al. (2020).

The simulations presented in Sections 5.1, 5.2, 5.3, 5.4, and 5.5 are based on a small urban scenario of the Prague–Holešovice crossroads of Dejvická and Komunardů streets, similar to the tiled base scenario used for the sensitivity study in Belda et al. (2020), which itself is based on the scenario used for validation of the PALM-USM model (Resler et al., 2017).

## 5.1 Convergence with respect to model resolution

The surface geometry and properties used in RTM are available with a certain level of detail and discretized by a regular grid. Hence a natural expectation would be that decreasing the grid spacing below certain level would not introduce new information and that the RTM model would converge to one solution. With RTM, increased model resolution leads to a higher number of finer faces and PCGBs. In order to investigate how sensitive the resulting radiative fluxes are on model resolution, multiple



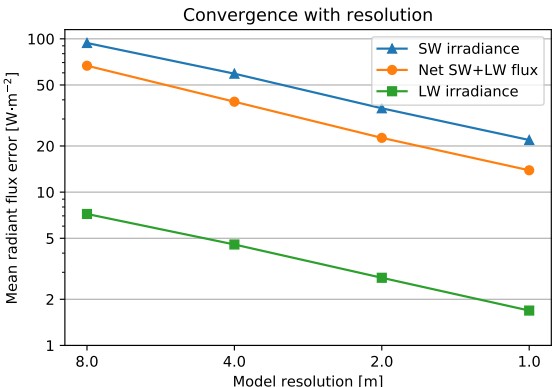

**Figure 6.** Double-logarithmic presentation of mean deviations of surface SW and LW irradiance as well as net radiant flux against the finest resolution case of 0.5 m.

simulations have been performed for the small urban scenario with resolution halved iteratively from 8 m down to 0.5 m. Because only radiative fluxes are of concern in this experiment, only one daytime timestep was compared.

The finest simulation with resolution of 0.5 m is taken as the base case and other scenarios are compared to it by radiative
fluxes at the matching surfaces. Finer resolutions mean increased detail in the 3-D structure of model surfaces, therefore not all surfaces represented in the finer resolution scenario correspond to the coarser-resolution scenario. In this experiment, around 70–80 % of fine resolution faces could be matched to respective coarse resolution faces. The results are shown in Figure 6.

On double-logarithmic scales the radiant flux errors decrease almost linearly. The largest errors can be observed in the SW fluxes with deviations to the reference case of almost $100\,\mathrm{W\,m^2}$, while errors in the LW fluxes are smaller by about one order
of magnitude. Extrapolating to even finer spatial resolution would imply that the mean error made for the LW fluxes become negligible, while the mean error for SW fluxes is still in the order of a few $\mathrm{W\,m^2}$. However, we emphasize that this is not related to the RTM itself but to the edged representation of sloped surface geometry on the Cartesian grid, which successively approaches the 'real-world' surface geometry with increasing spatial model resolution so that mutual surface reflections become more realistic.

**5.2 Convergence of angular discretization**

The angular resolution of the angular discretization scheme (see Section 2.2.4) can be controlled by setting the number of horizontal (azimuth) and vertical (elevation) directions. The default values are 80 and 40 steps respectively, i.e. 4.5° steps. This section explores the convergence of increasing angular resolution on the small urban scenario.

The angular discretization resolution also controls the discretization of direct solar irradiance, therefore different angular
resolutions also lead to a different number of discrete apparent solar positions throughout the day. For this experiment, a one day long simulation was performed with five different angular resolutions: 18°, 9°, 4.5°, 2.25° and 1.125°.



**Table 1.** Scaling of angular resoltution.

| Angular resolution (azimuth and zenith) | | 18° | 9° | 4.5° | 2.25° | 1.125° |
|---|---|---|---|---|---|---|
| Total discrete directions (hemisphere) | | 100 | 400 | 1,600 | 6,400 | 25,600 |
| Daily discrete solar positions | | 22 | 42 | 84 | 163 | 323 |
| VF entries | absolute (millions) | 1.8 | 5.3 | 13.4 | 26.2 | 40.2 |
| | per direction | 17,670 | 13,194 | 8,374 | 4,099 | 1,569 |
| CVF entries | absolute (millions) | 3.6 | 12.2 | 35.9 | 92.7 | 199.8 |
| | per direction | 35,971 | 30,523 | 22,409 | 14,482 | 7,804 |
| Raytracing calculation | [s] | 3 | 9 | 28 | 80 | 274 |
| RTM timestepping calculation | [s] | 50 | 139 | 378 | 812 | 1545 |

Table 1 lists parameters of the experiment. With doubling the angular resolution the number of discrete directions quadruples, while the number of view factors grows more slowly. This is a result of the aggregation of view factors with identical source and target face in the angular discretization scheme, as the increased angular resolution would surpass grid spacing for increasingly

distant mutually visible surfaces. For the target faces where the angular resolution is already finer than the grid spacing of the faces in its point of view, the increased number of rays traced from each visible face brings improved precision of the VF values without increasing the number of VFs.

The results for the convergence are shown in Figure 7, which shows mean absolute differences relative to the reference case with finest angular resolution (1.125°), for the selected radiative fluxes for each face throughout the 24-hour long experiment.

On double-logarihmic scales the mean deviation in the radiative fluxes are almost linear, with largest deviation again observed in the SW fluxes. However, compared to the grid resoultion, the error made by too coarse angular discretization is significantly smaller. Although increasing the angular resolution has relatively low demands on computational resources in comparison to increasing the spatial resolution of the model, the default value of 4.5° provides a reasonable tradeoff.

### 5.3  Convergence of multiple reflections

In order to quantify the appropriate number of reflections for typical urban scenarios, a simulation of the small urban scenario has been performed for one timestep of a summer daytime simulation with different number of reflection steps. To evaluate deviations in net radiant flux values, the reference scenario has been simulated with excessive 300 reflection steps, for which all remaining unreflected flux values have been almost zero, i.e. below the lowest positive value of the floating-point numerical representation.

The results are shown in figure 8. For SW radiation, the mean net radiant flux error is below $1\,\mathrm{W\,m^{-2}}$ after three reflections and the 95% quantile of the net flux error as well as the mean unreflected radiant flux is below $1\,\mathrm{W\cdot m^{-2}}$ after four reflections. For LW radiation, which reflects less in typical scenarios, these respective limits are reached one reflection earlier.





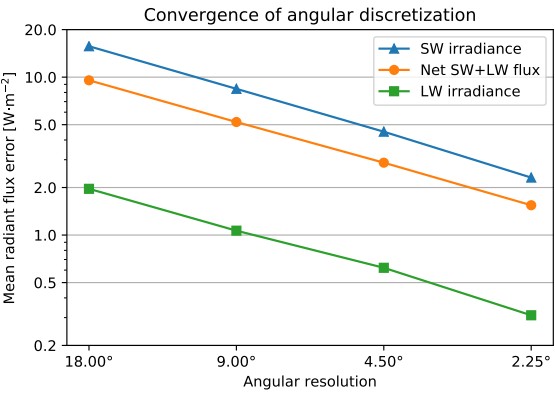

**Figure 7.** Double-logarithmic presentation of mean deviations of surface SW and LW irradiance as well as net radiant flux for different angular resolutions. Mean deviations are shown relative to the finest angular resolution of 1.125°.

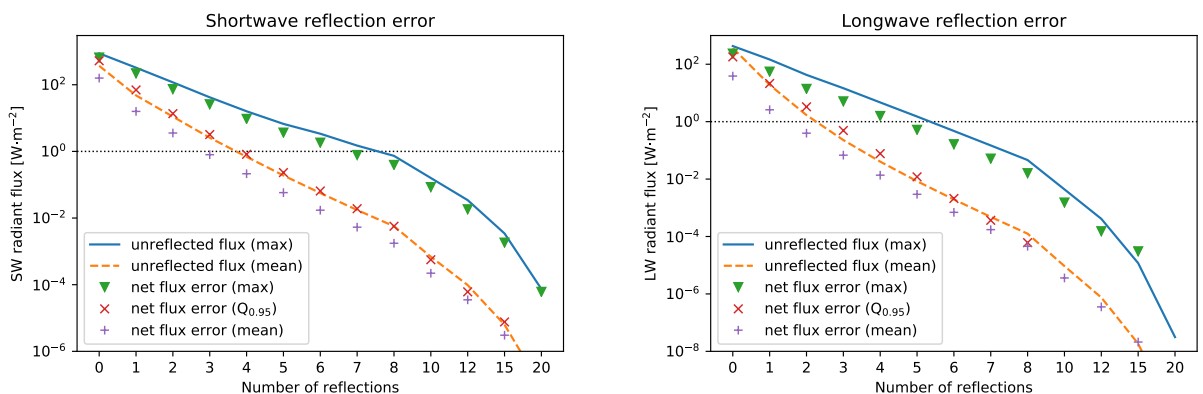

**Figure 8.** A double-logarithmic presentation of potential and actual errors in SW and LW radiation caused by insufficient number of reflections. The maximum and mean of the remainder of unreflected radiation per surface is is shown as lines, the absolute discrepancies of net radiant flux as compared to a perfectly reflected scenario is shown by individual points (maximum, 95th percentile and mean). The net flux errors above 15 reflections are zero (below the floating-point resolution), so are the 95th percentiles of LW error above 8 reflections.

These results support the recommendation to use the default RTM configuration value of three reflection steps for most scenarios. Considering that with the default radiation update interval of 60 s, the RTM uses only a small fraction of timestepping computational time, the number of reflection steps can be increased to e.g. five with negligible computational costs.





**Table 2.** Scaling of number of view factor entries.

| Horizontal grid size (repeated tiles) | | 1×1 | 2×2 | 4×4 | 8×8 | 16×16 |
|---|---|---|---|---|---|---|
| horizontal grid cells | thousands | 26 | 102 | 410 | 1,638 | 6,554 |
| VF entries | absolute (millions) | 13 | 59 | 250 | 1,025 | 4,152 |
| | per horizontal grid cell | 523 | 580 | 610 | 626 | 634 |
| CVF entries | absolute (millions) | 36 | 187 | 870 | 3,754 | 15,587 |
| | per horizontal grid cell | 1,401 | 1,825 | 2,125 | 2,291 | 2,378 |

## 5.4 Model scalability on large scenarios

To verify model scalability, a horizontal scaling experiment has been performed on the Salomon supercomputer at IT4Innovations National Supercomputing Centre[1]. The experiment was based on the small urban scenario of Prague–Holešovice.

The original model domain was doubled iteratively in both $x$- and $y$-direction, creating a tiled scenario with $2^n$ rows and $2^n$ columns ($2^{2n}$ copies) of the original domain for $n = 1 \ldots 4$. Each scenario was simulated using a proportional number of parallel processes, having 32 processes per tile and the total number quadrupling with each iteration. The Salomon supercomputer is composed of individual nodes with 24 CPU cores per node interconnected using the InfiniBand FDR fabric, therefore the scaling test used multiple nodes, testing also the scalability of remote data exchange.

For each domain size, a short 10-minute simulation was performed and the durations of individual tasks from model initialization and model timestepping were recorded together with the amount of view factor data entries as a measure of memory complexity. The radiation update interval was 60 s.

Table 2 lists the number of view factor entries for the horizontally tiled domains. Thanks to the constant maximum number of view factor entries in the angular discretization scheme, the actual number of entries per horizontal grid cell grows only slightly due to mutual visibility among the tiles, allowing less aggregation at more complex scenarios. The number of plant canopy view factor entries per face could grow proportionally to the mean ray length in the worst case, but the real case shows that, due to shading, the actual number of entries per face only increases moderately with exponentially larger domains.

The computational time measured by the scalability test is presented in Figure 9, split into the initialization phase (which is independent on the length of the simulation) and the actual timestepping phase. As can be seen in the log-log plot, the RTM initialization time (raytracing and data aggregation) is mostly proportional to the horizontal domain size, as expected from the increase in raytracing lengths and the amount of interprocess data exchange (see theoretical complexity in Section 3.2). Raytracing is the most data exchange intensive process in model initialization.

The temporal scaling of the timestepping phase of RTM is shown together with the timestepping of the rest of the model as a reference. RTM model calculation takes between 2–5 % of the timestepping phase. The largest simulated domain with 8,192 parallel processes running on 342 individual nodes displays slight worsening of the scaling curve both for RTM and for the rest

---

[1]https://www.it4i.cz





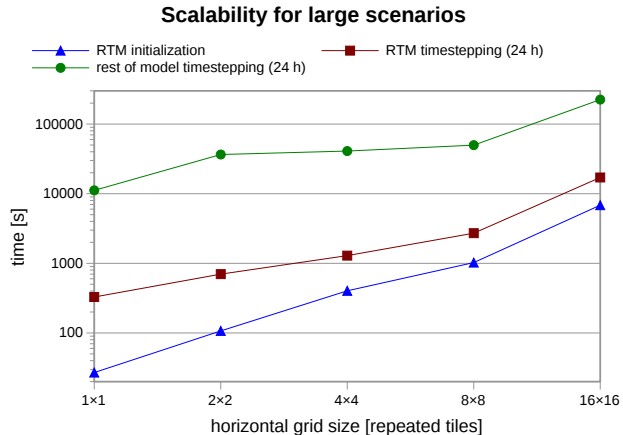

**Figure 9.** A double-logarithmic presentation of the computation time spent for different sub-tasks while simulating progressively larger domains (by the means of horizontal quadruplication). Each simulation uses a constant number of processes per horizontal tile. The sub-tasks shown are RTM initialization and timestepping and timestepping of the rest of the model as a reference. Timestepping time is shown for a 1-day long simulation as extrapolated from the 10-minute test simulations.

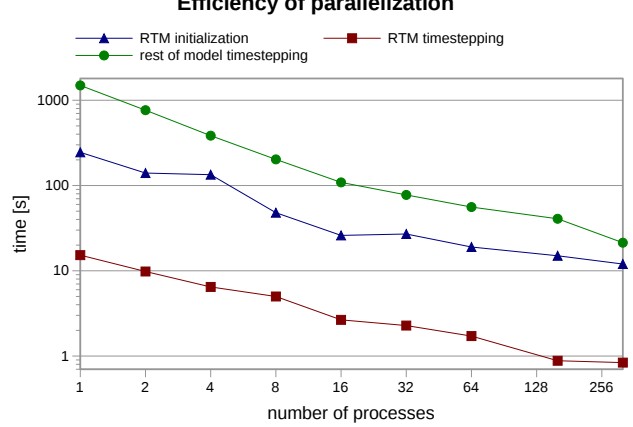

**Figure 10.** A double-logarithmic presentation of computational time versus the number of processes for a small scenario, typically suitable for 16–32 processes.

of the model, probably due to the growing complexity of interprocess data exchange. Future versions of RTM may be improved for the largest domains thanks to planned optimization of the amount of exchanged radiative flux data (see Section 6).



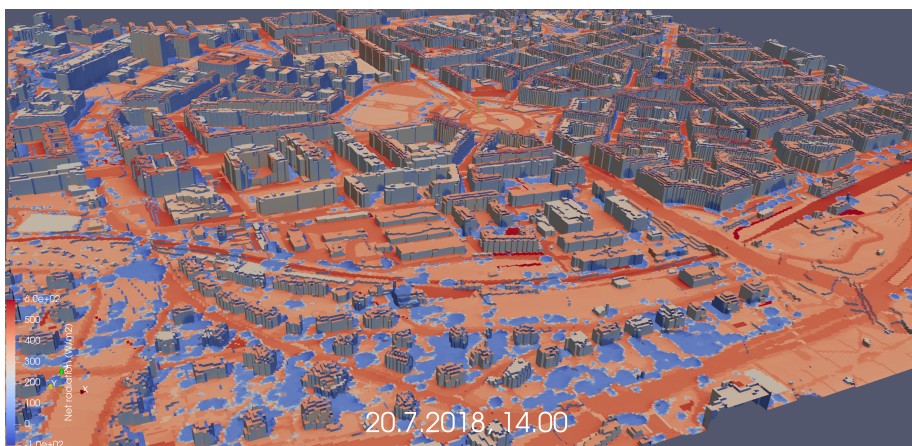

**Figure 11.** A 3-D representation of instantaneous net SW+LW radiative fluxes in a large urban scenario. A north-oriented view of the inner domain of the Prague–Dejvice validation scenario.

### 5.5 Efficiency of parallelization

Figure 10 shows the efficiency of parallelization for a small domain composed of a single tile of the scaling test domain. A domain of this size should be computed with up to 64 processors for reasonable simulation times, yet we explore an approx-imately exponential sequence starting with 1 process up to 320 processes, at which point each subdomain has only $10 \times 8$ horizontal grid cells.

We can see that between 1–16 processes the parallelization of both the initialization and timestepping phases is very good, even though the radiative interactions have very strong spatial interdependency, meaning significant mutual data exchange between subdomains. For further increasing number of processes, both the RTM initialization as well as RTM timestepping become less efficient. This is attributed to the relative increase of costs in MPI-communication compared to the cost of com-putations performed on each process, which is in accordance to Amdahl's law of strong scaling. In other words, when the subdomains become too small the speedup with increasing number of processes becomes less efficient.

### 5.6 Performance on large realistic urban scenario

Resler et al. (2020) focuses on validation of the PALM model on a large urban scenario, which is composed of two nested domains. The outer domain covers an area of 4 km × 4 km with a resolution of 10 m and the inner domain has an extent of 1440 m × 1440 m and a resolution of 2 m. An example of the simulated radiation for the inner domain is shown in Figure 11. The simulation was performed on a cluster with Infiniband EDR interconnection on 880 MPI processes. In this simulation, the RTM initialization took 10 minutes and within the 72-hour timestepping, the RTM calculation took between 0.5 % and 1.5 % of the time of computation, depending on meteorological conditions.



# 6    Conclusion and outlook

This paper gives a description of the currently implemented RTM in PALM. Also, sensitivity tests on performance-affecting configuration options (spatial model resolution, resolution of angular discretization and the number of reflection steps) are presented in this study, supporting their recommended configuration for typical urban scenarios.

Finally, the applicability of RTM on a large real-life scenarios is presented, demonstrating that the computational demands of RTM are in line with other components of the PALM model with respect to domain size. However, model validation on large scenarios and long-term experience with various realistic simulations have also identified specific weak points in model representativity and RTM's potential for further improvements (see Resler et al., 2020). New or improved simulated processes, different representation of model elements and possibilities for further improvements in computational efficiency and scalability are all included in upcoming development plans for the RTM.

## Fully three-dimensional buildings

Several modules in the PALM model, as well as the model core, now support fully 3-D structures with downward facing faces e.g. at bridges or lateral openings to courtyards. However, in many real-world scenarios overhanging structures are infrequent and only occur at a minor number of grid points. The current raytracing algorithms take advantage of the 2.5-D geometry to improve computational efficiency. Hence, the proposed update will still use the simplifications made for the 2.5-D geometry while enabling the fully 3-D support only at grid points where required.

## Immersed boundary method

For now, the representation of obstacles in PALM is fully based on the Cartesian grid, i.e. a grid box is either fully obstacle or fully atmosphere. As a consequence, surfaces that are actually slanted in reality, such as roofs or natural slopes, are represented as step-like surfaces. Beside its implications on the microscale flow biasing e.g. the surface friction, such step-like representation increases the total surface area in the model, which affects the amounts of radiative fluxes, and adds artificial shading and reflections. Future developments of PALM include the implementation of the *Immersed Boundary Method* (IBM) (Peskin, 1972) that allows to represent also slanted surfaces. This will allow e.g., to represent slanted roofs instead of step-wise roof-shapes, or a better representation of vertical buildings walls that are not perfectly aligned with the horizontal numerical grid. The implementation of IBM thus will also include changes in the raytracing algorithm in RTM where the surfaces may not necessarily be aligned parallel to the numerical grid axes.

## Specular reflections

All reflections are treated as Lambertian, i.e. fully diffuse, in the current version of RTM. Surfaces with mainly specular reflections, such as glass or polished metal surfaces, thus cannot be represented realistically. Multiple ways to implement specular reflections in RTM have been considered, but the feature would be of limited use with strictly Cartesian grid, therefore





the decision on how to implement specular reflections is being postponed after the implementation of immersed boundary
conditions.

### Localized raytracing

In the current parallelization of raytracing, the rays are traced as a whole in the process which owns the subdomain of the
ray's target. This has many computational advantages (see Section 3), but for very large domains with a lot of plant canopy
it leads to a large number of MPI calls, which can slow down raytracig substantially. Also, a proposed change for very large
domains where the global terrain elevation would not be copied to each process would further increase MPI communication in
raytracing.

A substantial change in raytracing algorithm is being considered, where each ray would be divided among segments be-
longing to individual subdomains. The process owning the subdomain of ray's target would successively ask the respective
processes that own other segments of the ray to perform the raytracing of those segments and it would aggregate the results.
However, this algorithm could be significantly slower for small domains. The advantages and disadvantages need to be verified,
and the new algorithm can be implemented as an option to the current raytracing algorithm, possibly with automatic switching.

### New discretization for direct irradiation of plant canopy

In the current implementation, the radiative fluxes absorbed in plant canopy are discretized differently for direct solar radiation
and for other radiative fluxes (diffuse, reflected and emitted radiation). Separate raytracing cycle needs to be performed to
calculate ray transmittances for discretized apparent solar positions for each PCGB and a sub-grid model is necessary for
direct solar irradiance (see Section 2.4.2), while for other fluxes only the rays to- and from faces are considered and no extra
raytracing is necessary.

The proposed change uses similar approach also for direct irradiance of plant canopy—using only the rays to- and from
905 faces. This approach has multiple benefits: it avoids the extra raytracing, which can take significant amount of time for large
domains with a lot of plant canopy, it unifies the discretization for all radiative fluxes in plant canopy and it guarantees that
the total plant canopy heat flux from direct solar irradiance equals the sum of irradiance deficit at surfaces caused by partial
shading from plant canopy. However, it neglects the absorbed fluxes from rays that would pass the domain without striking
any surface. This is only relevant for plant canopy near domain boundaries; on the other hand, such areas always suffer from
910 lack of radiative interaction with elements outside the domain and they cannot be considered representative anyway. Another
potential problem is a risk of moiré effect in the spatial distribution of plant canopy heat flux, which needs to be examined on
realistic scenarios.

### Optimized data exchange in timestepping

Current implementation of interprocess data exchange in timestepping uses MPI `gather` operation which distributes radiosi-
915 ties of all surfaces among all MPI processes. The `gather` operation takes advantage of tree topology exchange patterns in



modern MPI implementations (e.g. MVAPICH, Intel MPI) which makes it efficient and avoids complex data routing. The downside is increased memory complexity for very large scenarios (each process needs to hold arrays for all faces).

Two different approaches are currently being considered to improve scalability of this particular code. The first one takes advantage of the fact that typical simulations are performed on clusters with many CPU cores per node, where selected arrays can be allocated in shared memory with local access for all MPI processes running on the particular node, avoiding the need to allocate identical global arrays for each process and reducing intra-node communication.

The other considered approach involves creating a face visibility mapping among MPI processes, where each process allocates an array of visible faces from other subdomains, grouped and ordered by MPI process rank, and exchanging minimum amount radiosity data using MPI `alltoall` operation. The disadvantage of this approach is more complex data mapping and routing. The two proposed approaches need to be evaluated on different-sized domains and compared with the current implementation.

*Code availability.* RTM 3.0, as part of the PALM model, is a free software. Its source code is distributed under the GNU General Public License version 3[2] and it can be downloaded from the PALM website[3]. The code is managed in an SVN repository. The simulations presented in Section 5 were performed with SVN revision 4285 and the code of RTM, LSM, BSM, PCM, and BIO modules is available in supplements.

---

[2]https://www.gnu.org/licenses/gpl-3.0.html

[3]https://palm.muk.uni-hannover.de



## Appendix A: List of quantities

| Quantity | | Unit | Description |
|---|---|---|---|
| $\Phi$ | Radiant flux | W | Radiant power (energy per unit of time) of the respective process (emitted, reflected or absorbed by the described surface or object). |
| $J$ | Radiosity | $\mathrm{W\,m^{-2}}$ | Radiant flux leaving a surface per unit of area. |
| $E$ | Irradiance | $\mathrm{W\,m^{-2}}$ | Radiant flux received by a surface per unit of area. |
| $M$ | Radiant exitance | $\mathrm{W\,m^{-2}}$ | Thermal radiant flux emitted by a surface per unit of area. |
| $F_{i \to j}$ | VF | 1 | View factor from face $i$ towards face $j$. See definition in Section 2.1. |
| $F_i^{\mathrm{s}}$ | SVF | 1 | Sky view factor at face $i$. See definition in Section 2.4. |
| $F_{i,j}^{\mathrm{c}}$ | CVF | 1 | Canopy view factor for PCGB $i$ from face $j$. See definition in Section 2.3.1. |
| $T^{\mathrm{r}}$ | Ray transmittance | 1 | The ratio of the radiant flux transmitted (passed through) a partially transparent object to the radiant flux carried by the ray at the point where it enters the object. |
| $a$ | LAD | $\mathrm{m^{-1}}$ | Leaf area density. The ratio of total (one-sided) area of all plant leaves per unit of occupied volume. |
| $A$ | Area | $\mathrm{m^2}$ | |
| $T$ | Absolute temperature | K | |
| $\varepsilon$ | Emissivity | 1 | The ratio of LW radiation emitted or absorbed by a surface to that of an ideal black body. |
| $\sigma$ | Stefan–Boltzmann constant | | $\approx 5.67037 \times 10^{-8}\,\mathrm{W\,m^{-2}K^{-4}}$ |

*Author contributions.* PK and JR are the core authors of the methods, algorithms and implementation of the RTM, including the coupling to the BSM, PCM, and BIO modules. MS is the author of RTM integration to the PALM radiation module and the coupling to the BSM and LSM modules. SS and MHS created and validated the RTM coupling with the forcing radiation model. VF is the author of the evapotranspiration and latent heat flux model. All authors contributed to the text of the article, as well as to debugging, validation, and maintenance related to RTM.

*Competing interests.* The authors declare no competing interests.

*Acknowledgements.* Financial support was provided by the *Operational Program Prague – Growth Pole of the Czech Republic* project "Urbanization of weather forecast, air-quality prediction and climate scenarios for Prague" (CZ.07.1.02/0.0/0.0/16_040/0000383) which



is co-financed by the EU. The co-authors MS, SS and MHS were supported by the Federal German Ministry of Education and Research (BMBF) under grant 01LP1601 within the framework of *Research for Sustainable Development* (FONA)[4].

The simulations were performed on the HPC infrastructure of the Institute of Computer Science of the Czech Academy of Sciences (ICS) supported by the long-term strategic development financing of the ICS (RVO:67985807) and partly in the IT4I supercomputing centre which was supported by The Ministry of Education, Youth and Sports from the *Large Infrastructures for Research, Experimental Development and*

*Innovations* project "IT4Innovations National Supercomputing Center — LM2015070".

---

[4]https://www.fona.de



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
