# Peer review of "Radiative Transfer Model 3.0 integrated into the PALM model system 6.0"

_Geoscientific Model Development, 2020_

## Referee Comment (RC1) · Anonymous Referee #1 · 13 Sep 2020

This submission describes a number of reasonable and at times elegant developments in the PALM-RTM model to include vegetation and improve the computational efficiency and therefore feasibility for larger or more complex domains. In general the explanations are clear and the assumptions robust. This is a long and detailed submission treating a complex set of problems, and I commend the authors in general for this work. However, improvement is possible. In general, the methods section appears to draw on previous work in a number of instances but does not often make explicit reference to previous work. Second, treatment of shortwave radiation interactions with the plant canopy may omit important processes related to lower absorptivity of leaves in this radiation band, and I think further consideration could be given here. Third, some of the methods section might better be included in an Appendix or Supplementary to

keep the paper more focused and hold it to a more reasonable length.

Abstract and conclusions (Sect. 6): More details should be included. Which new processes, discretization schemes? What, specifically, are the core aspects of this submission that are novel?

Lines 17-20: Urban climate models embedded in mesoscale models typically do include radiation exchange in 2-D, as well as shading and multiple reflections.

Line 84: "around the vertical axis"

Line 87: Explain what the "f-plane" approach is, and give a reference.

Line 101: So the geometry is plane parallel.

Line 122: I don't think this is the case. Please explain.

Line 130: Reflections from plant canopies may substantially affect nearby surfaces/pedestrians.

Line 135-136: What about latent heat flux from leaves?

Lines 171-176: What happened to radiation intercepted by the plant canopy?

Line 178: Which simplifications? Exact calculation for plane parallel rectangles does not require performing any integration.

Line 206-218: But how high can the resolution go and still be computationally feasible?

Line 230: What about the 3rd dimension (e.g., especially in a neighbourhood with tall buildings)? Wouldn't it grow faster if vertical discretization also increased?

Lines 341-343: This is not necessarily a small assumption in the shortwave, where leaf scattering (reflection + transmission) is on the order of 50%. There is evidence that it is an important process.

Sect. 2.3: In general, which assumptions are made (or user controls are available)
related to orientation of leaves? In general in this section, I think additional references backing the decisions being made and relations used would be useful.

Line 352-354: Doesn't the attenuation depend on the direction from which the ray originates, and whether it intersects a small fraction of the PCGB at its edge, or right through the middle of the PCGB?

Eq. 8: Technically, this is the fraction of radiation that is not intercepted by a leaf. In reality, a good fraction of the radiation intercepted by leaves is transmitted forward as well.

Line 361: This assumption will overestimate the absorption of shortwave radiation by leaves, since a large fraction of shortwave radiation incident on leaves is reflected or transmitted through the leaves. The extinction coefficient can be modified to begin to take this fact into account.

Line 364-371: Doesn't this depend on the direction from which the ray originates? Or this is only used for shortwave interactions between faces (not for incident direct shortwave)? -> This appears to be answered in Sect. 2.4.1. Two points: 1) perhaps make this clear here (i.e., lines 364-371), and 2) won't this advance ray tracing of all solar position (line 425) incur a large penalty in terms of memory requirements? In general, it would be helpful if it were stated earlier, e.g. in Sect. 2.3, that treatment of direct shortwave radiation would come later (in Sect. 2.4).

Line 417: "an" -> "and"

Sect. 2.4.2: What about scattering (reflection & transmission) of intercepted direct solar radiation?

Line 482: What shape is this object?

Sect. 3 (and some of Sect. 4): Some of these sections may be better in an Appendix or Supplementary. These are key details, largely related to the computational approach, that are of interest to a select few. In other words, the *main* section of this submission

is overly focused on the computational details, given that it is a geoscientific modelling journal. It is already a very long and detailed paper.

Sect. 4.3: This is an approximate approach for stomatal conductance, best used at the landscape scale. It misses the nonlinear effects of solar irradiance distribution across the vegetation canopy. A better approach at this scale might be the two-big-leaf approach.

Fig. 11: Why are walls grey? They have SW+LW exchange, do they not?

Sect. 6: It could be made more clear that the bolded subheadings are current limitations and/or planned future developments. "New discretization for direct irradiation of plant canopy" – The difference between these two methods is not entirely clear based on this description.

---

## Referee Comment (RC2) · Anonymous Referee #2 · 11 Dec 2020

This submission describes a new verison of Radiative Transfer Model that integrated in PALM. The Radiative Transfer Model has been improved over previous versions, providing a more realistic representation of a wider range of urban scenarios. From my point of view, this paper is more like a report rather than a scientific article. It Describes the model (section 1-4). The originality of the paper needs to be further clarified. It seems the original contribution is only shown in section 5, in which a sensitivity study has been conducted. Beside, some points need to be clearfy, in order to improve the paper further: Line 177-179: How do it simplified the discretization of other processes? What are the other processes? Do the author make imporvment here? Line 450-454: Why does 0.9 be chosen? Is there any reference?

[Figure]

2020.

---

## Author Comment (AC1) · 5 Feb 2021

We would like to thank the reviewer for their detailed and insightful comments. We studied each comment carefully and we did our best to reply to each of them and to follow the suggestions. These comments and suggestions proved very helpful in improving the clarity and the scientific focus of the manuscript and we believe that the revised version has gained a lot in these aspects. On top of that, some of the reviewer's comments and suggestions provided valuable ideas that we will be considering in the future development of the RTM model. We follow with detailed replies to each individual comment.

*This submission describes a number of reasonable and at times elegant developments in the PALM-RTM model to include vegetation and improve the computational efficiency and therefore feasibility for larger or more complex domains. In general the explanations are clear and the assumptions robust. This is a long and detailed submission treating a complex set of problems, and I commend the authors in general for this work. However, improvement is possible.*

*In general, the methods section appears to draw on previous work in a number of instances but does not often make explicit reference to previous work.*
> We agree that the references were not complete. We tried to strengthen the manuscript in this respect during our revision and we supplied additional references and gave a better context for our work and its innovative aspects.

*Second, treatment of shortwave radiation interactions with the plant canopy may omit important processes related to lower absorptivity of leaves in this radiation band, and I think further consideration could be given here.*
> We agree that the treatment of SW interaction with plant canopy has its limitations and we have extended the discussion about the limitations in the appropriate parts of the manuscript. We also discuss our position on the importance of the respective processes in each specific comment related to this topic below.

*Third, some of the methods section might better be included in an Appendix or Supplementary to keep the paper more focused and hold it to a more reasonable length.*

> Although we had already made a significant effort to make the discussion version of the paper shorter, we agree with this suggestion. For the revised version, we have moved the Sections 3.1 and 3.3 (as numbered in the discussion version), which describe the implementation of the parts of RTM that have not changed significantly since RTM version 1.0 (which is described in Resler et al., 2017), to the Supplements. We also hope that this step will strengthen the scientific focus of the manuscript and make it less focused on the technical aspects.

*Abstract and conclusions (Sect. 6): More details should be included. Which new processes, discretization schemes? What, specifically, are the core aspects of this submission that are novel?*

> We agree that the novel methods and algorithms were not presented as such in the manuscript. We have updated the abstract, introductions of several sections and the conclusion so that the novel aspects are more clearly marked. We have also added a few additional references which better set this work into the context of related research.

*Lines 17-20: Urban climate models embedded in mesoscale models typically do include radiation exchange in 2-D, as well as shading and multiple reflections.*

> Indeed, yet the horizontal exchange is either neglected or crudely parametrized in mesoscale models due to the statistical nature of the description of the subgrid level topography and its parameters. The core novel technique here is the inclusion of explicit 3-D, fully resolved interaction. We agree that the formulation was misleading and we reformulated the paragraph and added further references.

*Line 84: "around the vertical axis"*

> Corrected.

*Line 87: Explain what the "f-plane" approach is, and give a reference.*

> The f-plane approach means that constant Coriolis parameters are used along the model extent to represent the Coriolis force, see e.g. the AMS glossary (https://glossary.ametsoc.org/wiki/F-plane_approximation). This approach is a valid assumption as long as the north-south model extension (along latitude) is in the order of a few or a few tens (even a few hundreds) of kilometers, which is the typical order of simulation setups with LES. We admit that the term should have been explained and we have added the explanation.

*Line 101: So the geometry is plane parallel.*

> Accepted, added to the text.

*Line 122: I don't think this is the case. Please explain.*

> This formulation was not quite clear. Its meaning was that without the addition of arbitrarily oriented faces, directional reflection would only be realistic for faces that are parallel to one of the grid planes. Radiation reflected directionally from a wrongly oriented face would cause overestimation of irradiance at the incorrectly simulated target as well as underestimation at the correct (realistic) target. This approach could therefore increase the error in the distribution of the reflected radiation, possibly even worse than the currently simulated Lambertian reflection. We thus decided to postpone the implementation of specular reflections after the implementation of the arbitrary oriented surfaces. As PALM is regularly used for cities with arbitrarily oriented streets and building walls (e.g. most of the European cities), the problem of the gridding of non-grid-aligned surfaces becomes a practical issue. We have reformulated the sentence to make its meaning more clear.

*Line 130: Reflections from plant canopies may substantially affect nearby surfaces/pedestrians.*

> We agree that this limitation might have a significant effect in specific cases. On the other hand, according to our experience with the 3-D structures of real urban scenarios, we suppose that the frequency of these cases where the amount of reflected radiation striking a nearby surface differs significantly from the background radiation is very low. We have added a forward reference to the discussion about this phenomenon in Section 2.3 (-> 2.4 in revised) and we have extended this discussion, acknowledging that this might be a limiting factor of applicability of the model for specific scenarios. We will also consider possible improvements in this regard for the future versions of RTM, with respect to the effect on model performance and scalability.

*Line 135-136: What about latent heat flux from leaves?*

> Latent heat flux is modelled using the coupled plant canopy module, as described in Section 4.3. The sentence has been reformulated to mention this.

*Lines 171-176: What happened to radiation intercepted by the plant canopy?*

> Radiation intercepted by the plant canopy is absorbed and converted into sensible and latent heat flux with the help of plant canopy sink/view factors, as described in Section 2.3.1 (-> 2.4.1 in revised). Section 2.1.1 briefly introduces basic view factor concepts and for clarity it does not deal with the absorbed flux yet. We considered adding a forward reference mentioning the absorbed flux, but we think that this would negatively affect the readability of Section 2.1.1.

*Line 178: Which simplifications? Exact calculation for plane parallel rectangles does not require performing any integration.*

> The integral could be solved numerically as part of the computation, which is not necessary indeed (the sentence was not meant to refer to a numerical solution). It could also be solved analytically for a list of categories depending on the

mutual position of the faces and the parametrized solution would then be applied in the model. The latter approach is used e.g. in Krayenhoff and Voogt (2007). We have considered this approach during the development of RTM 1.0, but due to interaction with resolved plant canopy with arbitrarily distributed LAD, this approach was deemed unavailable. Instead, we have chosen a simplified, yet more universal approach, which is described in the paper. The full integral and its simplification is presented in Section 2.1 (Eq. 2) and 2.2.1 (Eq. 4). Later development of the angular discretization method significantly decreased the error caused by this simplification. To make the statement in the manuscript more clear, we have added a reference to the integral.

*Line 206-218: But how high can the resolution go and still be computationally feasible?*
The scalability of increasing the resolution is described in the following section 2.2.2. Also a case study about convergence of increased resolution is presented in Section 5.1 (albeit for the angular discretization, yet that also uses singular rays). The sentence has been reformulated and the references have been added.

*Line 230: What about the 3rd dimension (e.g., especially in a neighbourhood with tall buildings)? Wouldn't it grow faster if vertical discretization also increased?*
The paragraph was not formulated clearly with respect to the third dimension. In case of increasing resolution uniformly in all three dimensions, the number of faces (which represent 2-D surfaces) grows with $x^2$. The paragraph has been reformulated.

*Lines 341-343: This is not necessarily a small assumption in the shortwave, where leaf scattering (reflection + transmission) is on the order of 50%. There is evidence that it is an important process.*
It is true that the shortwave reflection plays an important role in the plant canopy and this was not discussed sufficiently in the manuscript. In the sub-grid scale, multiple reflections increase the total transmission on top of the transmission through the leaves, and the scattering effect alters the direction of the radiation. The increase in transmissivity is taken into account in the value of the extinction coefficient α, which partially solves the problem for cases where the direction of radiation is not so important (e.g. shade under rather homogeneous layer of plant canopy or situations where the majority of plant canopy irradiance is not directional). The cases where the direction of radiation reflected from plant canopy may play an important role are discussed above in the reply for comment for line 130. We have added this discussion to the referred location in the manuscript and we have also added a paragraph in the Section 2.3.1 (-> 2.4.1 in revised) where we discuss the sub-grid effects. As mentioned in the reply above, we will consider improving the simulation of these effects in the future versions of RTM.

*Sect. 2.3: In general, which assumptions are made (or user controls are available) related to orientation of leaves? In general in this section, I think additional references backing the decisions being made and relations used would be useful.*

In general, the radiation properties and transpiration of a tree depend on the total leaf area, on the shape of the crown, on the distribution of the leaf area density (LAD) within the crown and also on the individual leaf orientation (both azimuth and inclination). The leaf orientation can depend, among others, on the direction of the incoming radiation and on the wind speed and direction, which change in time and individual leaves may also fluctuate in turbulence. The current model does not consider such details and the only user-controllable variable is the spatial distribution of LAD, which is discretized into individual grid cells. Within each discrete grid cell, LAD is assumed to be constant and the attenuation is homogeneous and isotropic. Leaves are assumed to be randomly oriented and the effects of variable leaf orientation are not considered. The average absorbed radiation per unit leaf area (based on LAD) is used for further computation of transpiration. It is assumed that the absorbed radiation flux per leaf area is constant in a grid cell.

In literature, there are different attitudes considering leaf orientation:
- Wang and Jarvis (1990, Tree Physiology 7) conclude for spruce: "(…)
  3. the LAD distribution within a tree crown has a large influence on the daily amounts of PAR absorbed by the crown of a tree in the stand;
  4. the average leaf incidence angle (LIA) has only a small influence on the daily amounts of PAR absorbed and transpiration, but a large influence on the daily amount of photosynthesis by the crown of a tree in the stand"
- James and Bell (2000, Tree Physiology 20) show how the leaf azimuth orientation facing east/west can help to increase radiation intercepted in the morning and the evening and reduce the leaf temperature at noon for Eucalyptus trees.
- Pisek et al. (2013, Agricultural and Forest Meteorology 169) state that while both the few existing field measurements and modeling results question the common practice of assuming the spherical leaf angle distribution and the authors themselves recommend the planophile or plagiophile leaf angle distribution, spherical distribution of leaf normals is often assumed.

Although we believe that this amount of details is out of scope of our paper considering the uncertainties present, we do agree that the attitude towards leaf orientation has not been described adequately. We have added a description of our modelling approach together with discussion and reference to the Section 2.3.1 (-> 2.4.1 in revised). In the future, we will consider adding some non-spherical leaf inclination distribution if very detailed simulations are desired. We have also added this possible improvement in the Outlook (Section 6.1).

*Line 352-354: Doesn't the attenuation depend on the direction from which the ray originates, and whether it intersects a small fraction of the PCGB at its edge, or right through the middle of the PCGB?*

> RTM assumes that the leaves are randomly oriented, so that the absorption is isotropic. The attenuation does indeed depend on the length of the intersection of the ray and the PCGB (which itself depends on the orientation and position of the ray with respect to the centre of the PCGB). This length is included in the formula (Eq. 8) and mentioned in the text following the formula. We agree that this has not been stated clearly, so we have extended the description of the length of the intersection. We have also added information about the isotropic absorption together with discussion and a reference in Section 2.3.1 (-> 2.4.1 in revised). See also our reply for comment on Section 2.3 above.

*Eq. 8: Technically, this is the fraction of radiation that is not intercepted by a leaf. In reality, a good fraction of the radiation intercepted by leaves is transmitted forward as well.*
*Line 361: This assumption will overestimate the absorption of shortwave radiation by leaves, since a large fraction of shortwave radiation incident on leaves is reflected or transmitted through the leaves. The extinction coefficient can be modified to begin to take this fact into account.*

> The formulation was not clear enough. By "the extinction coefficient α, which converts LAD (...) to a corresponding average optical density" we meant that the extinction coefficient is already specified in such a way that the absorption is not overestimated, and in Eq. 8 we mean a general exponential decrease in transmittance, which combines (approximately) the effects of un-intercepted leaves, leaf transmission and reflection. We have added a new paragraph that describes the meaning of Eq. 8, explains this modelling approach and cites the original source on which this approach was based in the (already cited) earlier versions of PALM. See also our replies to comments for line 130 and lines 341-343 above.

*Line 364-371: Doesn't this depend on the direction from which the ray originates? Or this is only used for shortwave interactions between faces (not for incident direct shortwave)? -> This appears to be answered in Sect. 2.4.1. Two points: 1) perhaps make this clear here (i.e., lines 364-371), (…)*

> We apply the same attenuation formula for all rays in the model and we assume that the leaves are randomly oriented, so that within each discretized PCGB the attenuation is homogeneous and isotropic. We agree that this has not been stated clearly. We have added a description and discussion about this in Section 2.3.1 (-> 2.4.1 in revised) and a reference explaining this modelling approach. See also our reply for comment on Section 2.3 above.

*(…) and 2) won't this advance ray tracing of all solar position (line 425) incur a large penalty in terms of memory requirements?*

The memory penalty from raytracing of solar positions is not large: for each face, there may be at most *n* view factor entries from *n* discretized directions. If the sun could shine from any direction, we would use at most *n* further factors for direct irradiance (because the angular discretization for radiation from surfaces is the same and it is actually computed together with almost no penalty on raytracing time), but even in long simulations, the discretized apparent solar positions form only a small fraction of all discretized directions, so the memory requirements for direct irradiance are much smaller than for the face-to-face view factors. We have reformulated the sentence to make this more clear.

*In general, it would be helpful if it were stated earlier, e.g. in Sect. 2.3, that treatment of direct shortwave radiation would come later (in Sect. 2.4).*

The reviewer has a good point about the flow of the text. We have added a statement in the Section 2.3 (-> 2.4 in revised) about the types of rays that are concerned, but on top of that, we have actually moved the (original) Sections 2.4 and 2.4.1 about the direct and diffuse solar radiation in front of the (original) Section 2.3 about the plant canopy and adapted them accordingly. We think that this order is more comprehensible, as in the original order it would be necessary to add many forward references about solar radiation to the section about plant canopy, while in the revised order there are only two forward references from solar radiation to plant canopy. We also think that it is now easier to understand the model of solar radiation without detailed knowledge about the modelled plant canopy than vice versa.

*Line 417: "an" -> "and"*

Probably refers to the line 407. The typo has been corrected.

*Sect. 2.4.2: What about scattering (reflection & transmission) of intercepted direct solar radiation?*

The effects of reflection and transmission of direct solar radiation in plant canopy are treated identically as for other types of radiation. The formula for absorption (Eq. 8 in Section 2.3.1 -> 2.4.1 in revised) uses the extinction coefficient α which takes into account the fact that transmitted radiation is higher than only the sum of rays that are not intercepted (see also replies above for comments on Eq. 8 and Line 361). This formula is used for attenuation of all traced rays, including the rays for direct solar radiation (Section 2.4.1 -> 2.3.1 in revised). The same formula is used for the sub-grid model of absorption of direct solar radiation in plant canopy (Section 2.4.2 -> 2.4.3 in revised), therefore the amount of absorbed radiative flux corresponds to the attenuated radiative flux from the specific rays. It is true that with the direct solar radiation, as opposed to diffuse and reflected radiation, the radiation scattered by plant canopy into different directions than the original rays has a greater potential to affect nearby surfaces significantly. The current approach is a result of a design choice with respect to computational complexity. It can be viewed as a limitation of the applicability of the model, which

needs to be considered for individual scenarios, as is discussed in the reply for the comment on line 130 above. We have extended the discussion of this issue in Section 2.3 (-> 2.4 in revised).

*Line 482: What shape is this object?*

The shape is configurable by the user, the basic shapes (sphere, ellipsoid, and simplified human body) are available as the configuration choices and other arbitrary shapes are easily extensible in the code. Details about the shape are described in Section 4.4. We have reformulated the forward reference and added a citation to an article with a more detailed description and specific formulae for the available shapes.

*Sect. 3 (and some of Sect. 4): Some of these sections may be better in an Appendix or Supplementary. These are key details, largely related to the computational approach, that are of interest to a select few. In other words, the \*main\* section of this submission is overly focused on the computational details, given that it is a geoscientific modelling journal. It is already a very long and detailed paper.*

We agree with the suggestion and we have moved the Sections 3.1 and 3.3 (as numbered in the discussion version), which describe the implementation of the parts of RTM that have not changed significantly since RTM 1.0, to the Supplements. On the other hand, the computational feasibility is one of the crucial problems of explicit 3-D modelling of radiative interactions. The innovative efficient implementation of RTM that ultimately allowed such explicit modelling consists not only of the underlying principles and algorithms, but also of the informatics aspect. For that reason, we consider the condensed description of computational approaches given in the revised Section 3 in accordance with the focus of GMD.

*Sect. 4.3: This is an approximate approach for stomatal conductance, best used at the landscape scale. It misses the nonlinear effects of solar irradiance distribution across the vegetation canopy. A better approach at this scale might be the two-big-leaf approach.*

We believe that the distribution within the canopy can be computed, provided the grid resolution is sufficient. Each resolved vegetation grid cell has a different amount of SW radiation absorbed, which is a result of the explicitly modelled 3-D radiation and also the discretized distribution of LAD. In a more refined approach, this could be distributed non-homogeneously even in the grid cells (based on the attenuation model), but in this version a single homogeneous leaf irradiation value is assumed within the grid cell. The distribution within a single tree can still be modelled in explicitly based on grid resolution and the availability of leaf distribution data. This approach could be indeed regarded as one-big-leaf, but per grid cell, not per one vegetation element, so the sunlit grid cells are shading other grid cells instead of having shaded and unshaded leaves in the two-big-leaf model. This is the first version of the resolved canopy transpiration parametrization and more refined parametrizations could be used in future

versions. Currently there are many other uncertainties present. For instance, there is a real lack of data for actual transpiration response of various common trees to environmental variables (PAR, air temperature, air humidity) and to the internal water pressure (which itself has a degree of uncertainty in the model). We agree that the shading effects have not been described clearly in the section and we have added a more clear explanation of the explicit approach to resolved plant canopy.

*Fig. 11: Why are walls grey? They have SW+LW exchange, do they not?*

The walls do have radiative exchange indeed and it is rendered in the figure, but the values of the net radiative flux were near the middle of the color mapping used in the figure, which made a lot of walls have gray shades. Also, the 3-D figure used shading, which made some walls darker and the values harder to distinguish. The figure has been re-rendered with a different color mapping and rendering parameters to make the values clearer to see. Please also note that the figure is intended rather as an illustration of the global result.

*Sect. 6: It could be made more clear that the bolded subheadings are current limitations and/or planned future developments.*

The section was not entirely clear and it has been slightly reorganized so that the future developments are more clearly marked as such.

*"New discretization for direct irradiation of plant canopy" – The difference between these two methods is not entirely clear based on this description.*

We agree that the methods were not described clearly and we tried to reformulate the descriptions. We have also added a remark about possible improvement with respect to non-circular leaf orientation distribution.

---

## Author Comment (AC2) · 5 Feb 2021

We would like to thank the reviewer for their valuable comments. We studied each comment carefully and we did our best to answer all the raised questions and to follow the suggestions. We follow with detailed replies to each individual comment.

*This submission describes a new verison of Radiative Transfer Model that integrated in PALM. The Radiative Transfer Model has been improved over previous versions, providing a more realistic representation of a wider range of urban scenarios.*

*From my point of view, this paper is more like a report rather than a scientific article. It Describes the model (section 1-4).*

> We believe that the structure and content of the manuscript, which thoroughly describes the redesigned RTM model, its implementation and the underlying principles, matches well its assigned manuscript type "Model description paper" in the journal Geoscientific Model Development, and that it fulfills the requirements for the manuscript type that are specified in https://www.geoscientific-model-development.net/about/manuscript_types.html#item1. It also fits the scope of the special issue "The PALM model system 6.0 for atmospheric and oceanic boundary-layer flows: model description and applications in urban environments" (https://gmd.copernicus.org/articles/special_issue999.html). Nevertheless, we tried to strengthen the scientific focus of the article and to reduce the amount of technical details, so the Sections 3.1 and 3.3 (as originally numbered) have been moved to Supplements. We also tried to increase the focus on the specification of the novel contributions in the manuscript.

*The originality of the paper needs to be further clarified. It seems the original contribution is only shown in section 5, in which a sensitivity study has been conducted.*

> We believe that the main original contribution is the authorship of the described model itself and the new features of the model presented in the manuscript, as is specified in "Author contribution" (line 532). We agree that the novel methods and algorithms presented in the manuscript have not been clearly marked as such in

some parts of the paper. To correct that, we have updated the abstract, several section introductions and the conclusion to make this specification more explicit.

*Beside, some points need to be clearfy, in order to improve the paper further:*

*Line 177-179: How do it simplified the discretization of other processes? What are the other processes? Do the author make imporvment here?*

The simplified calculation of view factors values is described in detail in Sections 2.2.1 (legacy discretization scheme) and 2.2.5 (angular discretization scheme). The "accordance with other processes" is meant as a reference to other processes in PALM (unrelated to radiation), which are also subject to spatial discretization, i.e. which have their own discretization errors that can be reduced by increasing the resolution of the grid. This comparison is mentioned in Section 2.2.1. To make this statement more clear, we have reformulated the paragraph and added a forward reference.

*Line 450-454: Why does 0.9 be chosen? Is there any reference?*

We agree that the choice of the value was not explained properly. To address that, we have added a more detailed explanation of the sub-grid discretization with a reference simulation. However, because the details are rather technical, we have added the new section to the supplements (Section S2.1) instead of the main paper.

---

## Author Response (AR2)

**Author's response**

Dear Editor,

thank you for your checking of our revised manuscript and for your comments. We agree with your suggestions and we have implemented them in our minor revision. We follow with the replies to each of your comments:

*Abstract: the anonymous referee #1 requested more information to the abstract. I only see changes in wording but I do not see that this answers the request of the reviewer. Could you please include more novelty detail in the Abstract with a bit better description of the improvements and maybe even how much certain points have improved (significantly is mentioned but maybe even give some numbers).*

> We agree that the changes to the abstract did not go as far as the comment by Referee #1 suggested. To correct that, we have extended the description of the novelty. We also tried to further specify the improvement with numbers, but we realized that any short numerical description without adequate context would be misleading, and the context needed to properly qualify the numbers would exceed the acceptable length of the abstract. Therefore we decided to specify the improvement by describing a very large simulation, which is further described in Section 5.4, to illustrate the improved performance and scalability.

*There are missing equation numbers through the manuscript. Please add these.*

> We apologize for the missing numbers. Originally we had misunderstood the GMD guidelines by thinking that it only specified the format of the numbering, and we only numbered the equations that were referenced from other parts of the manuscript. We have now added numbers to all equations and we have also verified that all the cross-manuscript references match the format specified by the GMD guidelines.

*P16, L404: You write "We believe that this difference can only be significant for very special cases (e.g. low azimuth direct solar radiation reflected from treetops towards the tops of nearby buildings where such buildings do not shade the the treetops completely) and we have not encountered significant amounts of such cases in our urban simulations, but the frequency of these cases has not been studied in detail" but this comment is very specific to the particular case(s) you have examined. Answer to the referee request should be made more general and not just mentioning that you have not encountered such issues but rather providing information. This can likely bring uncertainty but even if you have not encountered large differences in your cases it does not mean that someone else could not encounter.*

> We agree that this statement did not describe the issue adequately. We have replaced the description with a more general description and we have also acknowledged that our current knowledge of the impact of the disregarded effect across generic simulations is limited. We have also added a remark in

Section 6.1 (Outlook) that the impact and possible improvements are being studied.

**List of changes in the manuscript**

The line numbers in this section refer to the provided author's track-changes file generated by latexdiff.

**L6–12.** Abstract reformulated and extended as suggested by the editor.
**L77.** The novel raytracing algorithm was missing from the list by mistake, it has been added.
**L412.** Description extended for clarity.
**L424–433.** Description reformulated and extended as suggested by the editor.
**L961–964.** Added a mention about further research on the issue described in L424–433.

Apart from the listed changes, all equations have been numbered, which is rendered in latexdiff as if the equation was changed itself. All cross-references have been reformatted to match the GMD guidelines.